# A dynamic interplay between chitin synthase and the proteins Expansion/Rebuf reveals that chitin polymerisation and translocation are uncoupled in *Drosophila*

**Ettore De Giorgio[1], Panagiotis Giannios[1,2], M. Lluisa Espinàs[1], Marta Llimargas[1] \***

**1** Institut de Biologia Molecular de Barcelona, IBMB-CSIC, Parc Científic de Barcelona, Barcelona, Spain,
**2** Institute for Research in Biomedicine (IRB Barcelona), The Barcelona Institute of Science and Technology (BIST), Barcelona, Spain

\* mlcbmc@ibmb.csic.es

**Data Availability Statement:** All relevant data are within the paper and its Supporting Information files.

## Abstract

Chitin is a highly abundant polymer in nature and a principal component of apical extracellular matrices in insects. In addition, chitin has proved to be an excellent biomaterial with multiple applications. In spite of its importance, the molecular mechanisms of chitin biosynthesis and chitin structural diversity are not fully elucidated yet. To investigate these issues, we use *Drosophila* as a model. We previously showed that chitin deposition in ectodermal tissues requires the concomitant activities of the chitin synthase enzyme Kkv and the functionally interchangeable proteins Exp and Reb. Exp/Reb are conserved proteins, but their mechanism of activity during chitin deposition has not been elucidated yet. Here, we carry out a cellular and molecular analysis of chitin deposition, and we show that chitin polymerisation and chitin translocation to the extracellular space are uncoupled. We find that Kkv activity in chitin translocation, but not in polymerisation, requires the activity of Exp/Reb, and in particular of its conserved Nα-MH2 domain. The activity of Kkv in chitin polymerisation and translocation correlate with Kkv subcellular localisation, and in absence of Kkv-mediated extracellular chitin deposition, chitin accumulates intracellularly as membrane-less punctae. Unexpectedly, we find that although Kkv and Exp/Reb display largely complementary patterns at the apical domain, Exp/Reb activity nonetheless regulates the topological distribution of Kkv at the apical membrane. We propose a model in which Exp/Reb regulate the organisation of Kkv complexes at the apical membrane, which, in turn, regulates the function of Kkv in extracellular chitin translocation.

## Introduction

Chitin, a polymer of UDP-N-acetylglucosamine (GlcNAc) monomers, is a principal component of the apical extracellular matrix in arthropods. Chitin has a recognised importance in physiology [1,2] but also as a biomaterial [3]. Chitin and its deacetylated form, chitosan, are

**Funding:** This work was supported by the Spanish Ministerio de Ciencia e Innovación (FPI Fellowship BES-2016-076723 to EDG and BFU-2015-68098-P and PGC2018-098449-B-I00 grants to ML). PG is a researcher in Prof. Jordi Casanova's lab funded by Spanish Ministerio de Ciencia e Innovación (PGC2018-094254-B-100 grant) and the CERCA Program of the Catalan Government. The funders had no role in study design, data collection and analysis, decision to publish, or preparation of the manuscript.

**Competing interests:** The authors have declared that no competing interests exist.

**Abbreviations:** aa, amino acid; CC, coiled-coil; CDF, cumulative distribution function; CES, cellulose synthase; CHS, chitin synthase; CM2, conserved motif 2; exp, *expansion*; GlcNAc, N-acetylglucosamine; HS, hyaluronane synthase; IntDen, integrated density; kkv, *krotzkopf verkehrt*; NND, nearest neighbour distance; reb, rebuf; SDI, spatial distribution index.

nontoxic and biodegradable biopolymers with numerous applications in many sectors such as biomedicine, biotechnology, water treatment, food, agriculture, veterinary, or cosmetics [4,5]. So far, the main commercial sources of chitin are crab and shrimp shells [6]. Chitin isolation and purification from these sources require several treatments to remove proteins, calcium carbonate, lipids, and pigments, and no standarised methods exist nowadays [6]. In addition, these treatments have many industrial drawbacks such as high energy consumption, long handling times, solvent waste, high environmental pollution, and high economical costs, among others [4]. The synthesis of chitin in vitro can represent a more ecological, efficient, and "green" method as an alternative to the chemical procedures. Thus, it is critical to understand the molecular mechanisms of chitin deposition for a streamlined chitin production for multiple applications.

In insects, chitin is found in ectodermal tissues, where it forms chito-protein cuticles, and in the gut, where it forms a Peritrophic Matrix. Chitin is deposited to the extracellular space by chitin synthases (CHS) enzymes, which belong to the family of β-glycosyltransferases, which also includes cellulose synthases (CES) and hyaluronane synthases (HS). Most insect species encode two CHS types, CHS-A, required for chitin deposition in epidermis, trachea, foregut, and hindgut, and CHS-B, required for chitin deposition in the midgut, as a principal component of the peritrophic matrix [1,2,7]. The exact mechanism of chitin deposition is not fully elucidated yet, but it is proposed to occur in consecutive steps: (1) polymerisation by the catalytic domain of CHS; (2) translocation through the CHS of the nascent polymer across the membrane and release into the extracellular space; and (3) spontaneous assembly of translocated polymers to form crystalline microfibrils [1,2,8–10].

In *Drosophila*, CHS-A is encoded by *krotzkopf verkehrt* (*kkv*), which is responsible for chitin deposition in ectodermal tissues [11,12]. But besides *kkv*, our previous work identified another function exerted by *expansion* (*exp*) and *rebuf* (*reb*) to be required for chitin deposition. Exp and Reb are two homologous proteins, containing a conserved Nα-MH2 domain, that serve the same function, as the presence of only one of them can promote chitin deposition. In the absence of *exp/reb*, no chitin is deposited in ectodermal tissues, in spite of the presence of *kkv*, indicating that this function is absolutely required. But most importantly, we found that *kkv* and *exp/reb* compose the minimal genetic network, which is not only required, but also sufficient for chitin deposition. Thus, the concomitant expression of the two activities, *kkv+exp/reb*, promotes increased chitin deposition in ectodermally derived tissues that normally deposit chitin, like the trachea, and ectopic chitin deposition in ectodermally derived tissues that normally do not deposit chitin, like the salivary glands [13]. In spite of the capital importance of the *exp/reb* function in chitin deposition, the mechanism of activity of *exp/reb* has not been identified yet, nor their putative relation/interactions with *kkv*.

In this work, we have investigated the cellular and molecular mechanisms of chitin deposition in *Drosophila* and the roles of *exp/reb* and *kkv* in the process. We have found that the activities of Kkv in chitin polymerisation and chitin translocation are uncoupled, and we propose that chitin translocation, but not chitin polymerisation, requires Exp/Reb activity. Our cellular analysis has revealed that when extracellular chitin deposition is prevented, Kkv-polymerised chitin accumulates in the cytoplasm as membrane-less punctae. In addition, we detected a clear correlation between Kkv function in chitin polymerisation and/or translocation and Kkv subcellular localisation. A molecular analysis of Exp/Reb and Kkv proteins, using a structure–function approach, revealed key functions of different conserved motifs of these proteins in chitin polymerisation and extracellular deposition and in protein subcellular localisation. A detailed analysis of the subcellular localisation of Exp/Reb and Kkv indicates that these proteins display a largely complementary pattern at the apical membrane. However, in spite of this complementary pattern, we find that Exp/Reb regulate the pattern of distribution

of Kkv protein at the apical membrane. Based on the current understanding of the activity of glycosyltransferases like CES and on the knowledge of *Drosophila* CHS activity, we propose a model in which Exp/Reb regulate chitin deposition by modulating the distribution and organisation of Kkv complexes at the apical membrane, which would regulate the capacity of Kkv to translocate and release chitin fibers extracellularly.

## Results

### 1. The activities of Kkv in chitin polymerisation and translocation are uncoupled, and Exp/Reb activity is specifically required for chitin translocation

Chitin deposition is proposed to occur in 3 consecutive steps: (1) chitin polymerisation by CHS; (2) translocation of the nascent polymer through a CHS chitin-translocating channel across the membrane and release into the extracellular space; and (3) spontaneous assembly of translocated polymers to form crystalline microfibrils. In addition, it has also been proposed that the chitin polymerisation and translocation steps are tightly coupled [1,2,8–10]. However, no experimental data are available on Kkv with respect to this model. We aimed to investigate this model by carrying out a molecular and cellular analysis of the roles of Kkv and Exp/Reb in chitin deposition.

We found that Kkv can promote extracellular chitin deposition (which requires chitin translocation) in ectodermal tissues only in combination with Exp/Reb activity. *kkv* overexpression in the tracheal system rescues the lack of chitin in *kkv* mutants [13] and shows a comparable pattern to endogenous Kkv (S1A–S1D Fig). *kkv* overexpression does not affect extracellular chitin deposition (which starts from stage 13 as in the wild type) or tracheal morphogenesis (Figs 1A, 1B and S1E; [13]). However, we detected the presence of intracellular chitin punctae at early stages (before stage 14) (Figs 1A' and S3), which indicates the ability of *kkv* to polymerise chitin. These intracellular chitin punctae disappeared from stage 14, when chitin is then deposited extracellularly, and were not detected at later stages (Fig 1B'). This switch from intracellular chitin to extracellular chitin perfectly correlates with the expression of *exp/ reb* [13], suggesting that *exp/reb* promote extracellular chitin deposition. In agreement with this, in *exp reb* mutants overexpressing *kkv*, we found intracellular chitin punctae until late stages and no extracellular chitin deposition (Figs 1C and S3; [13]). We also found intracellular chitin punctae and no extracellular chitin when we overexpressed *kkv* in salivary glands, which do not express *exp* and *reb* (Figs 1D and S3; [13]). In contrast, the concomitant overexpression of *kkv* and *exp/reb* anticipates and increases extracellular chitin deposition in the trachea (S1E and S1F Fig), which leads to tracheal morphogenetic defects (S1G, S1H and S3 Figs; [13]). In addition, *kkv* and *exp/reb* coexpression in salivary glands promotes chitin deposition in the lumen (S1I and S3 Figs; [13]). No intracellular punctae of chitin were detected under these conditions, suggesting that all chitin synthesised by Kkv is deposited extracellularly by Exp/ Reb activity. Thus, we propose that the functions of Kkv in chitin polymerisation and translocation are uncoupled and that Exp/Reb activity is required for chitin translocation and release to the extracellular space. In this context, the presence of intracellular chitin would reflect the activity of Kkv in chitin polymerisation that cannot be further processed and translocated due to the absence of Exp/Reb activity.

We note that our experimental approach (using CBP staining to visualise chitin) could not detect intracellular chitin punctae produced by endogenous *kkv* in the absence of *exp/reb* in trachea or salivary glands. We propose that endogenous levels of Kkv are limiting and cannot produce sufficient intracellular chitin that we can detect. Therefore, we used the overexpression of *kkv*, which behaves as the wild-type protein, to augment its activity.

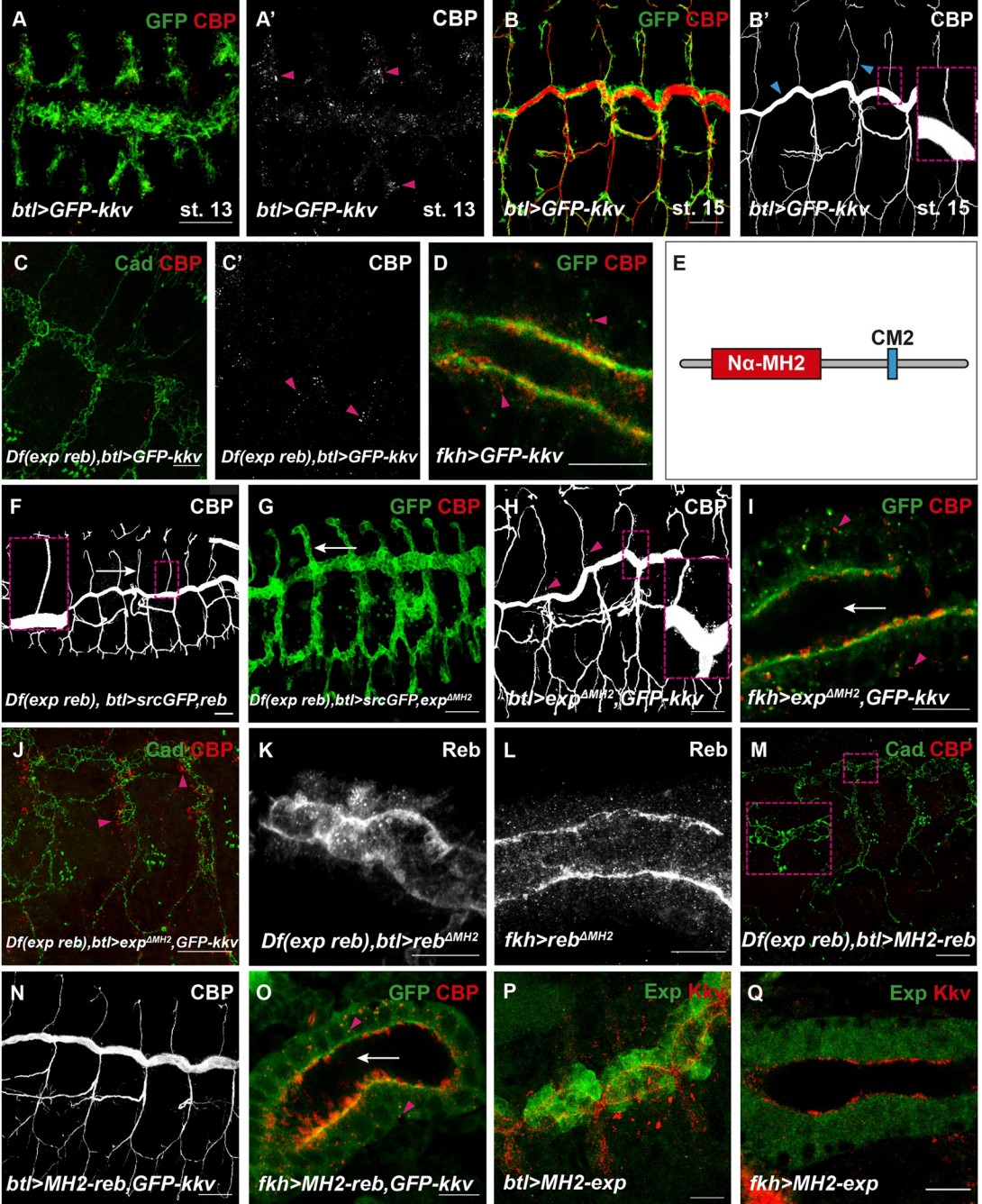

**Fig 1. Analysis of the role of the Nα-MH2 domain of Exp/Reb.** All images show projections of confocal sections, except D, I, L, O, and Q, which show single confocal sections. (**A**, **B**) The overexpression of *GFP-kkv* in the trachea leads to the presence of intracellular chitin vesicles at early stages (pink arrowheads) (**A, A'**). At later stages, intracellular chitin vesicles are not present, and chitin is deposited extracellularly in the lumen (blue arrowheads and inset) (**B, B'**). (**C, C'**) In *exp reb* mutants, the overexpression of *GFP-kkv* in the trachea produces intracellular chitin punctae until late stages (pink arrowheads). (**D**) The overexpression of *GFP-kkv* in salivary glands produces intracellular chitin vesicles (pink arrowheads). (**E**) Schematic representation of Exp protein. (**F, G**) In *exp reb* mutants, the expression of a wild-type form of *exp/reb* rescues the lack of extracellular chitin deposition (**F**, white arrow and inset), while *exp^{ΔMH2}/reb^{ΔMH2}* do not (**G**, white arrow points to absence of CBP). (**H, I**) The co-overexpression of *GFP-kkv* and *exp^{ΔMH2}* in control embryo does not produce morphogenetic defects in trachea (**H**) or extracellular chitin deposition in salivary glands (white arrow) (**I**); however, intracellular chitin punctae are present (pink arrowheads in **H, I**, and inset in **H**). (**J**) The coexpression of *GFP-kkv* and *exp^{ΔMH2}* in *exp reb* mutants produces intracellular chitin particles (pink arrowheads) but does not rescue the lack of extracellular chitin deposition. (**K, L**) Reb^{ΔMH2} localises apically in trachea (**K**) and in salivary glands (**L**). (**M**) *MH2-reb* is not able to rescue the absence of extracellular chitin

deposition in *exp reb* mutants. (**N, O**) The simultaneous expression of *MH2-reb* and *GFP-kkv* does not produce morphogenetic defects or ectopic chitin deposition in trachea (**N**) and in salivary glands (white arrow in **O**), but intracellular chitin vesicles are present (pink arrowhead in **O**). (**P, Q**) MH2-Exp protein does not localize apically in trachea (**P**) or in salivary gland (**Q**). Scale bars **A-C, F-H, J, M, N**: 25 μm; **D, I, K, L, O-Q**: 10 μm.

## 2. Structure–Function analysis of the roles of Exp/Reb in chitin translocation

We aimed to understand how Exp/Reb may regulate Kkv-dependent chitin translocation. The only recognisable domain of Exp/Reb identified was an Nα-MH2 [13,14]. However, in the course of this work, we identified a second domain highly conserved, which we called conserved motif 2 (CM2) (Fig 1E). We investigated the functional requirements of each of these two domains in chitin deposition. We generated different UAS transgenic mutant lines with the aim to evaluate their ability to rescue the lack of *exp reb* activity and their ability to promote chitin deposition when coexpressed with *kkv*. In agreement with their interchangeable activities, the results we obtained for *exp* or *reb* were comparable, so we will refer to them indistinctly.

### 2.1. Nα-MH2 domain is required for chitin translocation

We generated Exp and Reb mutant proteins that lacked the Nα-MH2 domain ($exp^{\Delta MH2}$, $reb^{\Delta MH2}$). The expression of full-length *exp* and *reb* rescues the lack of chitin deposition in the extracellular space in the trachea of *exp reb* mutants (Figs 1F and S3; [13]). In contrast, no extracellular chitin was deposited in an *exp reb* mutant background expressing $exp^{\Delta MH2}$ or $reb^{\Delta MH2}$ (Fig 1G), indicating that these proteins are not functional to promote chitin deposition.

In agreement with a role for the Nα-MH2 in promoting extracellular chitin deposition, we found that coexpression of *kkv* and $exp^{\Delta MH2}$/$reb^{\Delta MH2}$ did not lead to excessive chitin deposition and associated tracheal morphogenetic defects (Figs 1H and S3). We noticed, however, the presence of some intracellular chitin also at late stages, in contrast to the overexpression of *kkv* alone (Fig 1B). This result suggested that $exp^{\Delta MH2}$/$reb^{\Delta MH2}$ may interfere with endogenous *exp*/*reb*, maybe through interactions with other proteins in a complex (see Discussion). Coexpression of *kkv* and $exp^{\Delta MH2}$/$reb^{\Delta MH2}$ in salivary glands or in *exp reb* mutant trachea did not lead to ectopic chitin deposition (Fig 1I and 1J), as full-length *exp*/*reb* do (S1F, S1H, S1I, and S3 Figs; [13]). However, as expected, lack of extracellular chitin deposition was accompanied by the presence of intracellular chitin punctae (Figs 1I, 1J and S3). These results indicated that the activity of Exp/Reb in extracellular chitin deposition resides in its Nα-MH2, suggesting a role for the Nα-MH2 domain of Exp/Reb in chitin translocation and release to the extracellular space.

To discard an unspecific effect on extracellular chitin deposition due to the absence of the MH2 domain, we investigated $Exp^{\Delta MH2}$/$Reb^{\Delta MH2}$ localisation. We found that $Reb^{\Delta MH2}$ localised apically when expressed in the trachea (Fig 1K) or salivary glands (Fig 1L), in a comparable pattern to the overexpression of full-length Reb protein (S2A and S2B Fig). These results indicated that the Nα-MH2 domain is not required for Exp/Reb localisation and instead plays a specific role in extracellular chitin deposition.

As the Nα-MH2 is required for chitin extracellular deposition, we asked whether it was also sufficient to promote extracellular chitin deposition. We generated UAS constructs with only the Nα-MH2 domain of Exp and Reb (*MH2-exp* and *MH2-reb*). We found that the constructs were unable to rescue the lack of extracellular chitin deposition in *exp reb* mutants and to produce tracheal morphogenetic defects in the trachea or chitin luminal disposition in salivary

glands when coexpressed with *kkv* (Figs 1M–1O and S3). This indicated that the MH2-Exp/MH2-Reb proteins are not functional to promote extracellular chitin deposition. In agreement with this observation, we also detected the presence of numerous intracellular puncta of chitin (Figs 1O, S2C and S3). Correlating with this lack of activity, we observed that MH2-Exp did not localise apically, as Exp does (S2D Fig), and instead, it was found in the cytoplasm in the trachea or salivary glands (Fig 1P and 1Q). Our results suggested that either other domains of the protein also contribute to extracellular chitin deposition or that apical localisation is required for Exp/Reb activity. The results also indicated that other domains in Exp/Reb are required for apical localisation.

In summary, the analysis of the Nα-MH2 domain points that it is required for translocation and extracellular release of chitin but it is dispensable for protein localisation and chitin polymerisation.

## 2.2. The conserved motif 2 is required for Exp localisation

We searched for conserved domains by comparing the amino acid sequences of several Exp homologs. We identified a highly conserved region not previously described in the literature. The conserved motif 2 (from now on CM2) contained a region of 8 aa highly conserved (blue square in Fig 2A) followed by a region of 9 aa less conserved (red square in Fig 2A). To investigate the functional activity of the CM2, we generated $UASexp^{\Delta CM2}$ transgenic lines.

The expression of $exp^{\Delta CM2}$ in an *exp reb* mutant background rescued the lack of extracellular chitin deposition (Figs 2B and S3), indicating that the protein is functional. In agreement with this, when coexpressed with *kkv* in the trachea, it produced morphogenetic defects (Figs 2C and S3), comparable to the overexpression of *kkv* and *exp/reb* (S1H and S3 Figs). Similarly, expression of *kkv* and $exp^{\Delta CM2}$ in salivary glands produced ectopic chitin deposition in the luminal space (Fig 2D). These results indicated that the CM2 is not required for chitin polymerisation and translocation to the extracellular space. In agreement with no requirements of the CM2 in chitin translocation, no intracellular chitin punctae were detected when coexpressing *kkv* and $exp^{\Delta CM2}$ (Figs 2D and S3).

To further investigate the role of the CM2, we analysed protein localisation. The endogenous Exp and Reb proteins localise mainly apically at the membrane, although a bit of the protein can be detected intracellularly (S2D Fig; [13]). This pattern of subcellular accumulation was maintained when overexpressing Exp (Fig 2E). In contrast, overexpressed $Exp^{\Delta CM2}$ did not show such a distinct apical localisation (Fig 2F). We analysed the ratio of accumulation of the Exp proteins (control and $Exp^{\Delta CM2}$) in apical versus basal regions. Quantifications indicated a decreased apical enrichment of $Exp^{\Delta CM2}$ compared to full-length Exp (Fig 2H). This result indicated that the CM2 is involved in Exp/Reb localisation. However, we could still detect accumulation of $Exp^{\Delta CM2}$ at the apical membrane (Fig 2F), which correlated with its activity in chitin deposition.

In summary, we identified a highly conserved domain in Exp proteins that is dispensable for chitin polymerisation and translocation but is required for correct protein localisation, which we propose is important for Exp/Reb activity (see Discussion).

## 3. Structure–Function analysis of two conserved Kkv domains in chitin deposition

Kkv is a large protein with multiple functional domains and several transmembrane domains [9,15,16]. As for other members of the β-glycosyltransferase family, it is proposed that the activity of CHS, like Kkv, depends on oligomerisation and interactions with other proteins [1,2,9]. We investigated two Kkv domains with putative roles in direct or indirect interactions

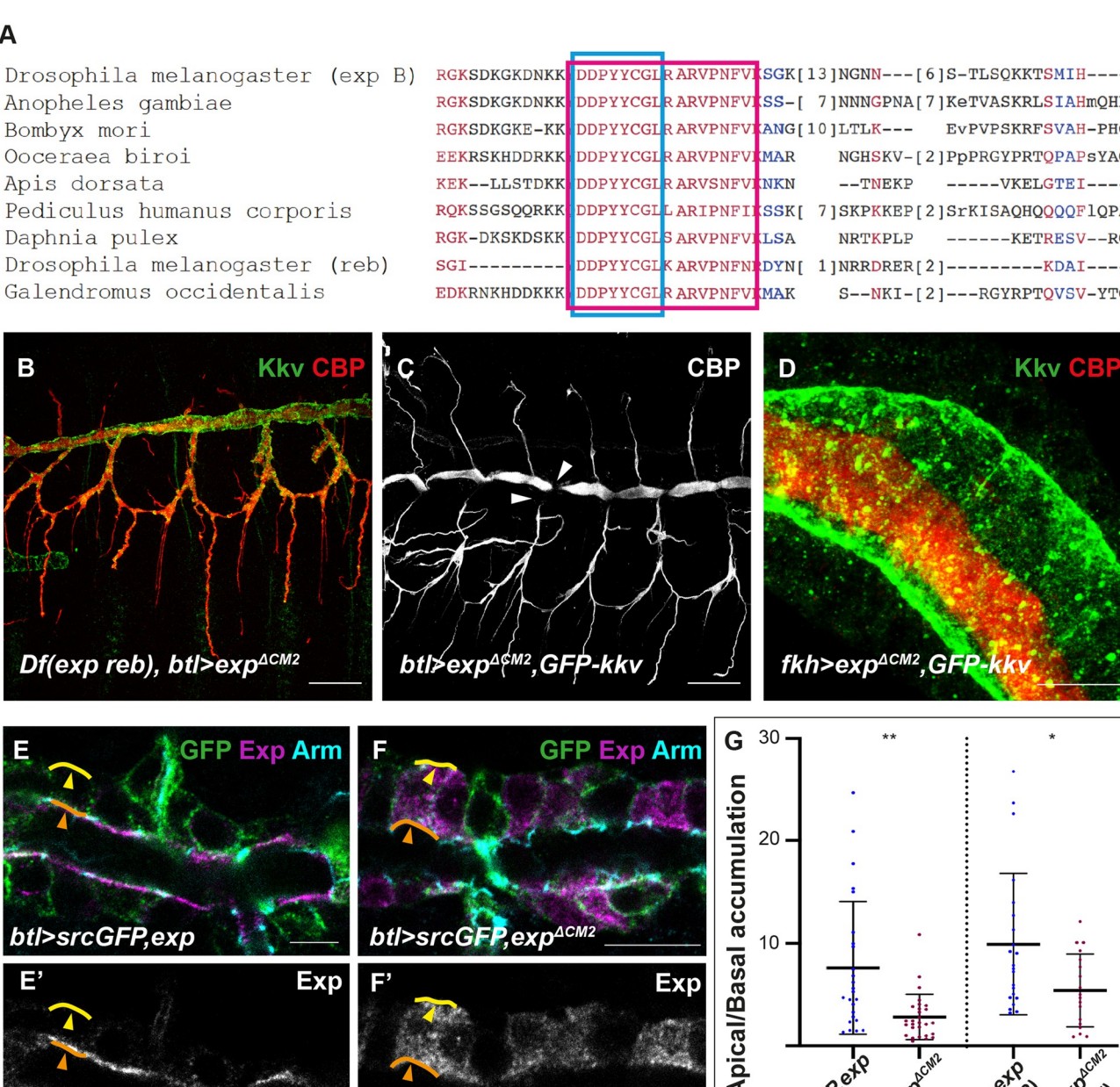

**Fig 2. Analysis of the role of the CM2 domain of Exp/Reb.** (**A**) Alignment of amino acids (aa) sequences of the isoform B of Exp (aa 356–433) and homologs; the blue square indicates 8 aa highly conserved, the red square includes 9 aa less conserved. (**B**-**D**) Show projections of confocal sections and (**E**-**F**) show single confocal sections. (**B**) The overexpression of $exp^{ΔCM2}$ in an *exp reb* mutant background rescues the lack of extracellular chitin deposition. (**C, D**) The simultaneous expression of $exp^{ΔCM2}$ and *GFP-kkv* produces morphogenetic defects in the trachea (arrowheads in **C**) and ectopic chitin deposition in the lumen of salivary glands (**D**). (**E, F**) Overexpressed Exp localises mainly in the apical region (orange arrowheads) with respect the basal domain (yellow arrowheads), while the apical accumulation of overexpressed $Exp^{ΔCM2}$ is less conspicuous (**F**). (**G**) Quantifications of accumulation of Exp and $Exp^{ΔCM2}$ in apical versus basal region. n corresponds to the number of ratios analysed (apical/basal ratio per cell), and brackets indicate the number of embryos used. Ratios were obtained from the apical (orange line in **E**) and basal (yellow line in **E**) domains of single cells in trachea and salivary glands. The underlying data for quantifications can be found in the S1 Data. Scale bars **B**, **C**: 25 μm; **D**-**F**: 10 μm.

with other proteins or in oligomerisation [11,15], the conserved motif WGTRE (amino acids (aa) 1,076–1,080) and a coiled-coil (CC) domain (aa 1,087–1,107) (Fig 3A).

### 3.1. The WGTRE domain is required for Kkv ER-exit

The conserved motif WGTRE was proposed to be an essential domain for Kkv activity as a point mutation changing the glycine renders an inactive protein [12]. This domain has been suggested to be involved in oligomerisation or interactions with other factors [11]. We generated a protein lacking this domain, GFP-Kkv$^{\Delta WGTRE}$.

To determine the activity of this protein, we assayed its rescuing capacity in a *kkv* mutant background. While a wild-type form of *kkv* can rescue the absence of chitin produced by the absence of *kkv* [13], *GFP-kkv$^{\Delta WGTRE}$* could not, and trachea was defective (Figs 3B and S3). This indicated that the protein is not functional, confirming that the WGTRE domain is essential for chitin production. Accordingly, expression of *GFP-kkv$^{\Delta WGTRE}$* did not produce chitin vesicles in the trachea at early stages (Figs 3C and S3), or in salivary glands (Fig 3D), as *GFP-kkv* does (Figs 1A, 1D and S3), indicating absence of chitin polymerisation.

To better understand the role of this domain, we analysed the localisation of the GFP-Kkv$^{\Delta WGTRE}$ protein. We found no apical accumulation of this protein, neither in a wild-type background nor in a *kkv* mutant background (Fig 3B). Instead, we found a generalised pattern in the cytoplasm. Costainings with the ER marker KDEL (Fig 3E) indicated that the GFP-Kkv$^{\Delta WGTRE}$ protein is retained in the ER.

ER retention may be due either to a defective folding of the protein or to a specific effect of this domain in Kkv trafficking to the membrane. Proteins with a defective folding are degraded from ER upon ubiquitination [17]. To distinguish between these two possibilities, we used the FK2 antibody that recognises mono- and polyubiquitinated conjugates, but not free ubiquitin [18]. From our results, GFP-Kkv$^{\Delta WGTRE}$ and FK2 do not colocalise (Fig 3F), indicating that the protein is not ubiquinated. The results strongly suggested a role for the WGTRE domain in Kkv trafficking from the ER to the membrane. We also concluded that ER exit is required for chitin polymerisation by Kkv.

### 3.2. The coiled-coil domain is required for Kkv localisation and full Kkv activity

Class A CHS contain a CC domain localised C-terminal to the active centre. Potentially, the CC domain could mediate association to yet unknown partner/s, or be involved in protein oligomerisation, regulating CHS localisation or activity [8,15,16].

We generated a protein lacking the CC region, GFP-Kkv$^{\Delta CC}$. This mutant protein rescued the lack of chitin in a *kkv* mutant background (Figs 3G and S3), indicating that it is functional. Accordingly, the concomitant expression of *reb* and *GFP-kkv$^{\Delta CC}$* in salivary glands resulted in deposition of chitin in the luminal space (Figs 3H and S3). Altogether, these results indicated that GFP-Kkv$^{\Delta CC}$ acts as a functional protein in these contexts.

We found, however, a condition in which *GFP-kkv$^{\Delta CC}$* behaved differently from *GFP-kkv*. The overexpression of *reb* and full-length *GFP-kkv* in the trachea produced strong morphogenetic defects (S1H and S3 Figs; [13]). In contrast, the overexpression of *reb* and *GFP-kkv$^{\Delta CC}$* did not produce this abnormal phenotype, and instead, tubes and chitin deposition appeared normal (Figs 3I and S3). Importantly, the overexpression of *reb* alone also leads to morphogenetic defects, due to the presence of endogenous *kkv* (Figs 3J and S3). Because the overexpression of *reb* and *GFP-kkv$^{\Delta CC}$* (in the presence of endogenous *kkv*) reverts the tracheal defects of overexpression of *reb* alone, our results suggest that the *GFP-kkv$^{\Delta CC}$* is interfering with the

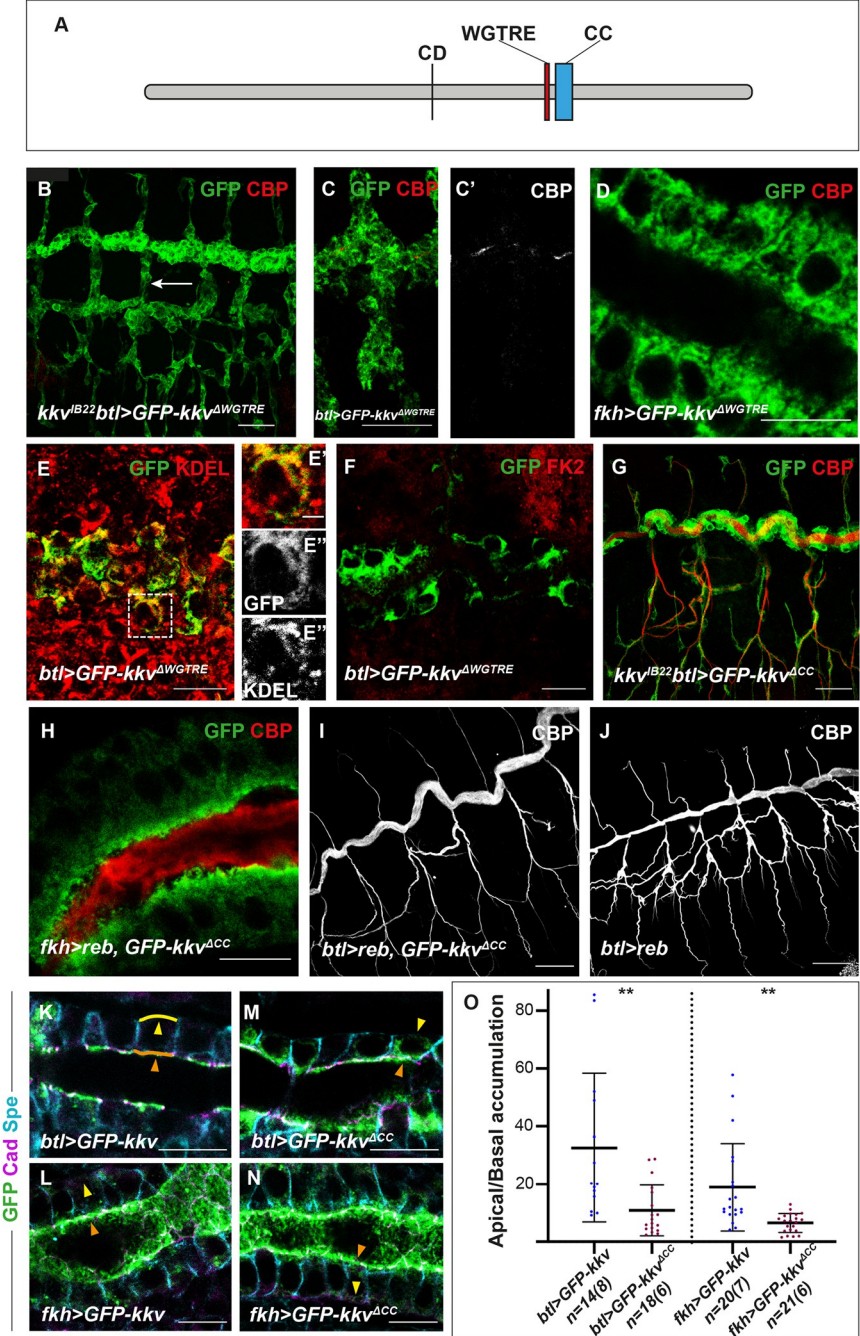

**Fig 3. Analysis of the WGTRE and CC domains of Kkv.** (**A**) Schematic representation of Kkv protein (CD, catalytic domain; WGTRE; CC, coiled-coil domain). (**B, C, G, I, J**) Show projections of confocal sections and (**D-F, H, K-N**) show single confocal sections. (**B**) The overexpression of *GFP-kkv$^{\Delta WGTRE}$* in a *kkv* mutant background does not rescue the absence of extracellular chitin deposition (white arrow, note the absence of CBP) and the protein accumulates in a generalised pattern. (**C-D**) The overexpression of *GFP-Kkv$^{\Delta WGTRE}$* does not produce intracellular chitin punctae, neither in trachea at early stages (**C-C'**) nor in salivary glands (**D**). (**E-E'''**) GFP-kkv$^{\Delta WGTRE}$ colocalise with the ER marker KDEL. (**F**) GFP-Kkv$^{\Delta WGTRE}$ does not colocalise with the marker FK2. (**G**) The overexpression of *GFP-kkv$^{\Delta CC}$* in a *kkv* mutant background rescues the lack of extracellular chitin deposition in the trachea (note the presence of CBP staining). (**H, I**) The simultaneous expression of *reb* and *GFP-kkv$^{\Delta CC}$* in salivary glands produces ectopic extracellular chitin (**H**), and no defects in trachea (**I**). (**J**) The overexpression of *reb* in trachea leads to morphogenetic defects. (**K, L**) Overexpressed GFP-Kkv localises mainly apically (orange arrowheads) although a bit of the protein can be detected in the basal region (yellow arrowheads). (**M, N**) Apical accumulation of overexpressed GFP-Kkv$^{\Delta CC}$ is less conspicuous. (**O**) Quantifications of accumulation of GFP-Kkv and GFP-Kkv$^{\Delta CC}$ in apical versus basal region. n corresponds to the

number of ratios analysed (apical/basal ratio per cell), and brackets indicate the number of embryos used. Ratios were obtained from the apical (orange line in **K**) and basal (yellow line in **K**) domains of single cells in trachea and salivary glands. The underlying data for quantifications can be found in the S1 Data. Scale bars **B**, **C**, **G**, **I**, **J**: 25 μm; **D-F**, **H**, **K-N**: 10 μm.

endogenous wild-type Kkv, which can no longer produce a dominant effect in combination with extra Reb.

We analysed the localisation of GFP-Kkv$^{\Delta CC}$ to further determine the roles of the CC domain. We found that the protein can still localise at the apical domain, as GFP-Kkv does (Fig 3K–3N); however, we found that the apical enrichment was not as clear as for GFP-Kkv. Quantification of the ratio of protein accumulation in the apical domain versus the basal domain indicated significant differences with respect the control. We confirmed these observations in salivary glands (Fig 3O).

Altogether, these results indicate that the CC domain is dispensable for polymerisation and translocation of chitin but plays a role in protein localisation. In addition, the results suggest that GFP-Kkv$^{\Delta CC}$ can interfere with endogenous Kkv, which may indicate a role in protein oligomerisation (see Discussion).

## 4. Cellular analysis of chitin polymerisation and extracellular translocation

We aimed to investigate further at the cellular level the roles of Kkv in chitin polymerisation and translocation and the nature of the intracellular chitin deposition. To this purpose, we used the salivary glands. Salivary glands express *kkv* (Figs 1Q and S4A) but do not express *exp* or *reb* and do not produce chitin. As shown previously, the coexpression of *kkv* and *exp/reb* in salivary glands leads to the deposition of chitin extracellularly in the lumen (Figs 4A and S1I; [13]). In contrast, in the absence of *exp/reb* activity, like when coexpressing *kkv* and *exp$^{\Delta MH2}$*/*reb$^{\Delta MH2}$*, we find intracellular chitin accumulation and no extracellular chitin in the lumen (Fig 4B and 4C). As indicated before, this suggested that chitin is deposited intracellularly because it cannot be translocated to the extracellular space. However, other mechanisms could explain intracellular chitin accumulation. For instance, intracellular accumulation could result from an abnormal endocytic uptake of previously translocated chitin occurring in the absence of *exp/reb* activity. To test this possibility, we blocked endocytosis in *GFP-kkv+exp$^{\Delta MH2}$*/*reb$^{\Delta MH2}$* expressing conditions. We still detected intracellular chitin punctae, indicating that these result from lack of chitin translocation (Fig 4D).

The use of the *UAS-GFP-kkv* line served us to overexpress *kkv*, but also to visualise *kkv* accumulation. Expression of *GFP-kkv* in salivary glands revealed the expected Kkv localisation at the apical membrane, but also the presence of many intracellular Kkv punctae in the cytoplasm. We asked whether intracellular chitin punctae and Kkv punctae colocalised. We found that around 34% of GFP-Kkv punctae partially colocalised with chitin, and around 20% of intracellular chitin punctae partially colocalised with GFP-Kkv (Fig 4B). We also found many examples in which GFP-Kkv and chitin punctae were in very close proximity (Fig 4B–4B"). Intracellular chitin punctae distributed throughout the cytoplasm, but we also detected large amounts of chitin deposits at the apical domain of the cell (Fig 4C). Nevertheless, orthogonal sections showed that this apical chitin is not deposited extracellularly in the lumen (Fig 4C'), but instead localised intracellularly in the apical region, in contrast to the coexpression GFP-Kkv and wild-type Reb (Fig 4A').

To identify the nature of Kkv and chitin punctae, we performed colocalisation analysis with different markers of intracellular trafficking. Colocalisation with the TransGolgi marker Golgin 245 indicated that several GFP-Kkv punctae corresponded to secretion vesicles (around

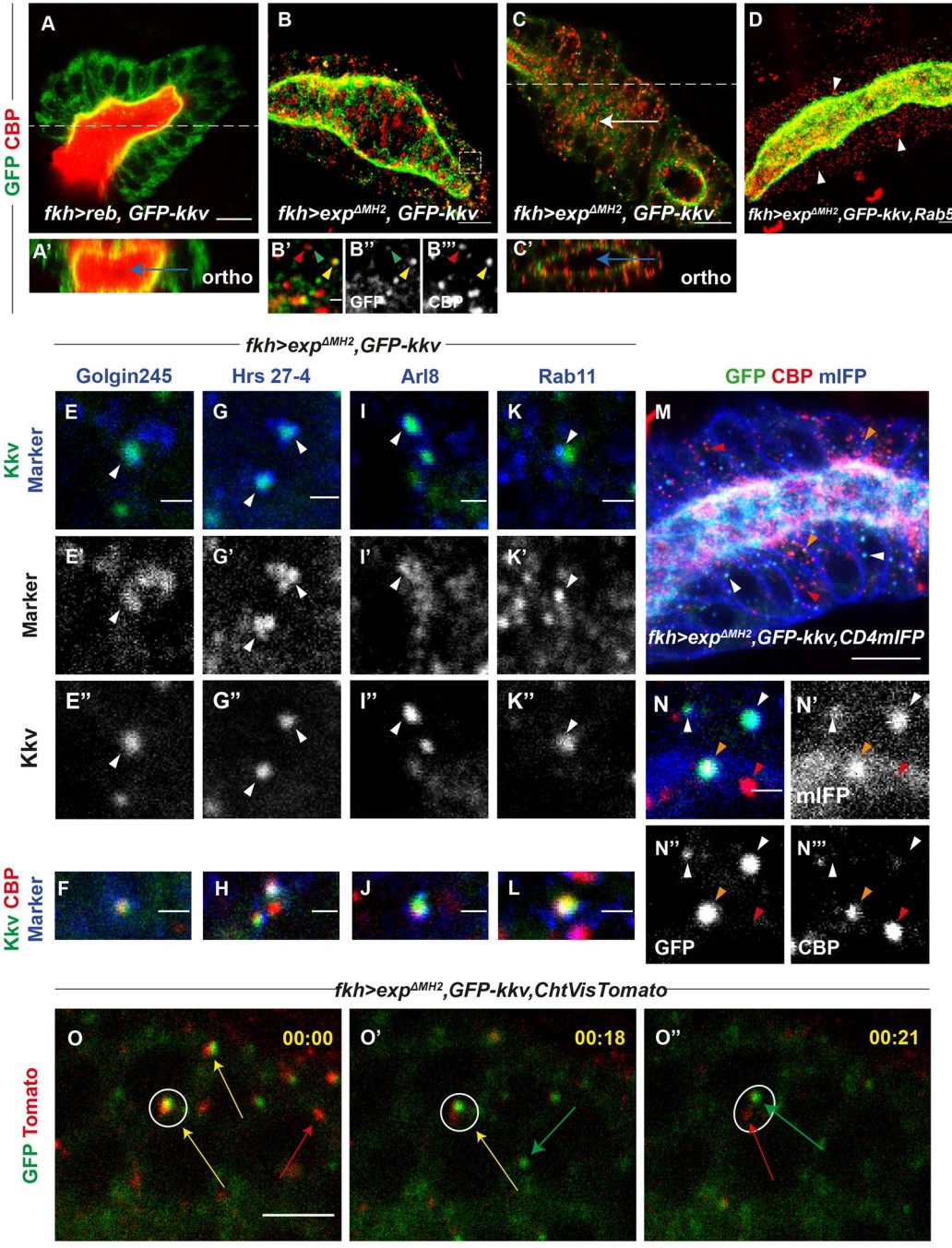

**Fig 4. Analysis of intracellular chitin deposition.** All images show salivary glands. (**A, C, E-L, N, O**) Show single confocal sections and (**B, D, M**) show projections of several sections. (**A-A'**) The concomitant expression of *reb* and *GFP-kkv* leads to luminal chitin deposition (blue arrow in orthogonal section in **A'**). (**B, C**) Coexpression of *exp^{ΔMH2}* and *GFP-kkv* produces intracellular chitin punctae, some of which partially colocalise with GFP-Kkv vesicles (yellow arrowhead) while others do not (red arrowhead). GFP-Kkv vesicles without chitin are also observed (green arrowhead). Note the accumulation of chitin in the apical domain (white arrow in **C**) that is not deposited extracellularly in the lumen (blue arrow in orthogonal section in **C'**). (**D**) In *Rab5^{DN}* background, intracellular chitin punctae are still present (white arrowheads). (**E-L**) Analysis of the nature of GFP-Kkv vesicles and chitin punctae using markers Golgin245 (**E-F**), Hrs 27–4 (**G-H**), Arl8 (**I-J**), and Rab11 (**K-L**); arrowheads indicate colocalisation between Kkv and each specific marker. (**M, N**) All GFP-Kkv vesicles colocalise with the membrane marker CD4-mIFP (white arrowheads), and few of them also with chitin (orange arrowheads); single chitin punctae do not colocalise with CD4-mIFP (red arrowheads). (**O-O''**) Frames from live imaging movie show that partially colocalising GFP-Kkv and chitin punctae (yellow arrow) can separate from each other; however, many GFP-Kkv (green arrow) and chitin puncta (red arrow) do not colocalise. Scale bars **A-D, M**: 10 μm; **E-L, N-N'''**: 1 μm; **O-O''**: 5 μm.

12% of GFP-Kkv vesicles) (Fig 4E). A few of these GFP-Kkv secretion vesicles partially colocalised with chitin (Fig 4F). We also detected that many GFP-Kkv vesicles colocalised with late endosomal markers (around 60% of GFP-Kkv vesicles), like the ESCRT-0 complex component Hrs, indicating an endocytic recycling of Kkv (Fig 4G). Some of these endocytic Kkv vesicles were also positive for chitin (Fig 4H). To investigate whether endocytosed Kkv was then following the degradation pathway or was recycled back to the membrane, we used lysosomal markers and markers for vesicle recycling. We found colocalisation of GFP-Kkv and the lysosomal marker Arl8 (35% of GFP-Kkv vesicles), indicating that part of the protein is degraded (Fig 4I and 4J). Finally, we also found colocalisation of GFP-Kkv with Rab11 (around 15% of GFP-Kkv vesicles), a marker for recycling (Fig 4K). Again, a few of them also partially colocalised with chitin (Fig 4L). Altogether, the results suggested that Kkv is transported to the membrane, internalised, and recycled back to the membrane or degraded. During this trafficking route, Kkv protein seem to be able to polymerise chitin.

We realised that many intracellular chitin punctae did not colocalise with GFP-Kkv or with any other trafficking marker, suggesting that they may not correspond to intracellular vesicles. To determine whether these chitin punctae accumulated in membrane-confined compartments, we used the general plasma membrane marker CD4::mIFP [19], which is enriched in plasma membrane and other subcellular membrane compartments (Fig 4M). We found that while all GFP-Kkv vesicles colocalised with the CD4::mIFP marker, the single chitin punctae did not (Fig 4N), indicating that these chitin punctae are not confined in membranous vesicles. Several proteins have been shown to bind chitin fibers modulating in this way the organisation and function of the chitinous apical extracellular matrix [20–22]. We asked whether the intracellular chitin punctae accumulated chitin-binding proteins. We observed that the chitin deacetylases vermiform and serpentine and the chitin-binding protein Gasp did not colocalise with intracellular chitin punctae (S4B–S4F Fig). Altogether, these results suggest that the intracellular chitin punctae correspond to short and naked chitin fibers floating freely in the cytoplasm.

To further understand the nature and dynamics of these chitin punctae, we performed live imaging using a live-probe for chitin and GFP-Kkv. The results confirmed that many GFP-Kkv and chitin punctae did not colocalise. Nevertheless, we found examples in which we detected GFP-Kkv and chitin partially colocalising and observed that KkvGFP and chitin were then separating from each other (Figs 4O and S4G and S1 and S2 Movies).

Altogether, our results are consistent with a model in which Kkv can polymerise chitin facing the cytoplasm in a constitutive manner, and when this chitin is not translocated, it accumulates intracellularly as short fibers (see Discussion).

## 5. Kkv activity correlates with Kkv trafficking and localisation

Previous experiments suggested that Kkv activity plays a role in Kkv localisation [16]. In wild-type embryos, endogenous Kkv protein (visualised with anti-Kkv) is found localised apically and in intracellular vesicles (Fig 5A), as it is the case of GFP-Kkv expressed in salivary glands (Fig 5C). We investigated whether this pattern of localisation was affected by Kkv activity.

We found that in conditions of lack of chitin polymerisation (i.e., in amorphic *kkv* mutants with a point mutation in the catalytic domain, *kkv*[63–20] mutants; [11]), Kkv did not localise apically, and instead, it was detected in the cytoplasm (Fig 5B). In conditions where chitin is polymerised but not translocated and accumulates in intracellular punctae (i.e., when expressing *GFP-kkv* in the absence of *exp/reb* activity), Kkv is secreted, reaches the membrane, is then internalised and degraded or recycled back to the membrane, as previously described (Figs 4 and 5C). In conditions where chitin is massively produced and deposited extracellularly (i.e.,

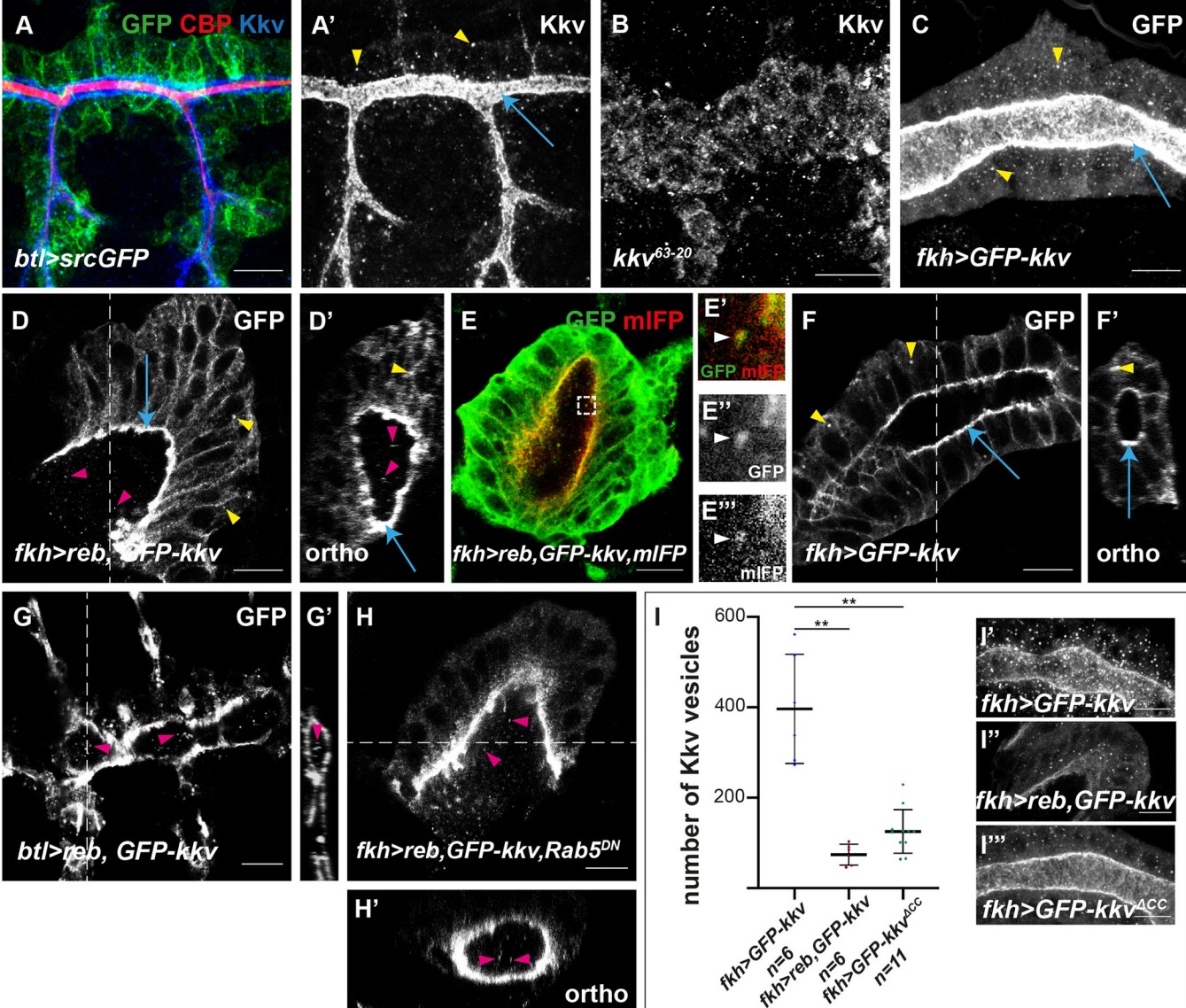

**Fig 5. Analysis of Kkv trafficking.** All images are single confocal sections except **A**-**C**, which are projections of confocal sections. (**A, A'**) In trachea of wild-type embryos, Kkv is present in the apical region (blue arrows) and in intracellular vesicles (yellow arrowheads). (**B**) In *kkv* mutants unable to polymerise chitin, Kkv is not properly localised. (**C**) GFP-Kkv localises to the apical region (blue arrows) and in intracellular vesicles (yellow arrowheads). (**D**) When *reb* and *GFP-kkv* are coexpressed in salivary glands, Kkv is present in the apical membrane (blue arrow), in intracellular vesicles (yellow arrowhead), and also in punctae in the lumen (pink arrowheads). This is clearly observed in orthogonal sections (**D'**). (**E**) These luminal punctae corresponded to membranous structures. (**F, F'**) In contrast, when *GFP-kkv* is expressed alone, luminal punctae are absent, and Kkv is only found apically (blue arrow) and in intracellular vesicles (yellow arrowhead). (**G**) Luminal punctae (pink arrowheads) are also observed in the trachea of embryos overexpressing *reb* and *GFP-kkv*. (**H**) When endocytosis is prevented, the coexpression of *reb* and *GFP-kkv* in salivary glands still leads to formation of Kkv luminal punctae (pink arrowheads). (**I**) Quantifications of the number of intracellular Kkv vesicles in salivary glands when expressing *GFP-kkv* (**I'**), *reb* and *GFP-kkv* (**I"**), and *GFP-kkv*^ΔCC. n is the number of salivary glands analysed. The underlying data for quantifications can be found in the S1 Data. Scale bars: 10 μm.

when expressing *reb* and *GFP-kkv* in salivary glands), we detected Kkv at the apical membrane and in vesicles (Fig 5D), although we detected less vesicles than with *GFP-kkv* alone (Figs 5C, 5I and S2E). In addition, we detected punctae of GFP-Kkv in the lumen of salivary glands (pink arrow in Fig 5D), which corresponded to membranous structures (Fig 5E), and which were not detected in conditions of GFP-Kkv expression alone (Fig 5F). These extracellular punctae, which were detected also by Kkv antibody (S4H Fig), correlate with a huge

production and extracellular deposition of chitin by *kkv*. In agreement with this observation, we also found GFP-Kkv punctae in the lumen of the trachea overexpressing *GFP-kkv* and *reb* (Fig 5G), which deposit increased amounts of chitin. These punctae seem similar to the extracellular Kkv punctae described in [16,23] and could therefore reflect Kkv shedding. Proteins can be shed to the extracellular space as extracellular vesicles, which comprise exosomes (derived from the endocytic trafficking) and microvesicles (directly shed from the plasma membrane) [24,25]. We could detect extracellular vesicles in salivary glands expressing *reb* +*GFP-kkv* in which we blocked the endocytic uptake, by expressing *Rab5^{DN}* (Fig 5H). This result suggested that the Kkv vesicles that we detect extracellularly may correspond to microvesicles (see Discussion).

Finally, we also found differences in Kkv localisation and trafficking when the protein lacks the CC domain (Fig 5I). We observed a reduced number of GFP-Kkv^{ΔCC} vesicles compared to GFP-Kkv. This result suggests that the CC domain is directly or indirectly involved in Kkv trafficking.

Altogether, our results point to a correlation between Kkv activity and Kkv trafficking and localisation.

## 6. Exp/Reb activity is required for Kkv apical distribution

Because we observed a correlation between Kkv activity and Kkv localisation, we asked whether *exp/reb* activity regulates Kkv localisation. We had previously shown that *exp/reb* activity is not required for apical localisation of overexpressed *GFP-kkv* [13]. However, the overexpression of *kkv* in this experimental setting could mask a possible role of *exp/reb* in Kkv localisation. Thus, we revisited this issue analysing the accumulation of endogenous Kkv using an antibody that we generated. We confirmed that Kkv localised apically in the absence of *exp/ reb* (Fig 6A and 6B). We found that Kkv also localised apically when expressing the nonfunctional proteins Exp^{ΔMH2} or MH2-exp in trachea (Fig 6C and 6D).

To investigate in detail possible subtle differences in Kkv apical accumulation, we used super-resolution microscopy and compared control embryos and *exp reb* mutants at different stages. At stage 14, both in control and *exp reb* mutant embryos, Kkv was found apical but also in many intracellular vesicles. Kkv at the apical membrane showed a nonuniform pattern in intensity and distribution (Fig 6E and 6F). At stage 16, control embryos showed an organised apical distribution pattern of Kkv in stripes, corresponding to the taenidial folds, and Kkv vesicles were largely absent (Fig 6G). In *exp reb* mutants at stage 16, Kkv was also apical, but in contrast to the control, Kkv did not distribute in stripes, and instead, we observed a nonuniform pattern of distribution (Fig 6H). Because in *exp reb* mutant conditions chitin is not deposited and taenidia do not form [13], this raised the possibility that the effects observed are due to the absence of taenidia. To discard this possibility, we analysed stage 15 embryos in which the chitinous cuticle forming the taenidia is not yet deposited but chitin is strongly deposited in the luminal filament. In control embryos, we did not observe a clear pattern of organisation for the apical Kkv, although its localisation was not visibly discontinuous and could be suggestive of some level of order (Fig 6I). In *exp reb* mutants, we observed a nonuniform distribution of apical Kkv protein with different intensities, resembling earlier stages (Fig 6J). To characterise further the topological distribution of Kkv on the apical area and identify possible alterations in *exp reb* mutants, we used spatial statistics to analyse our samples based on two main parameters; the distance between each Kkv punctum and its closest neighbour within a reference area defined by the apical marker Armadillo, and the distance between the puncta and arbitrary positions within the same reference area. Relevant to these parameters, two different cumulative distribution functions (CDFs) defined as G and F—Function were

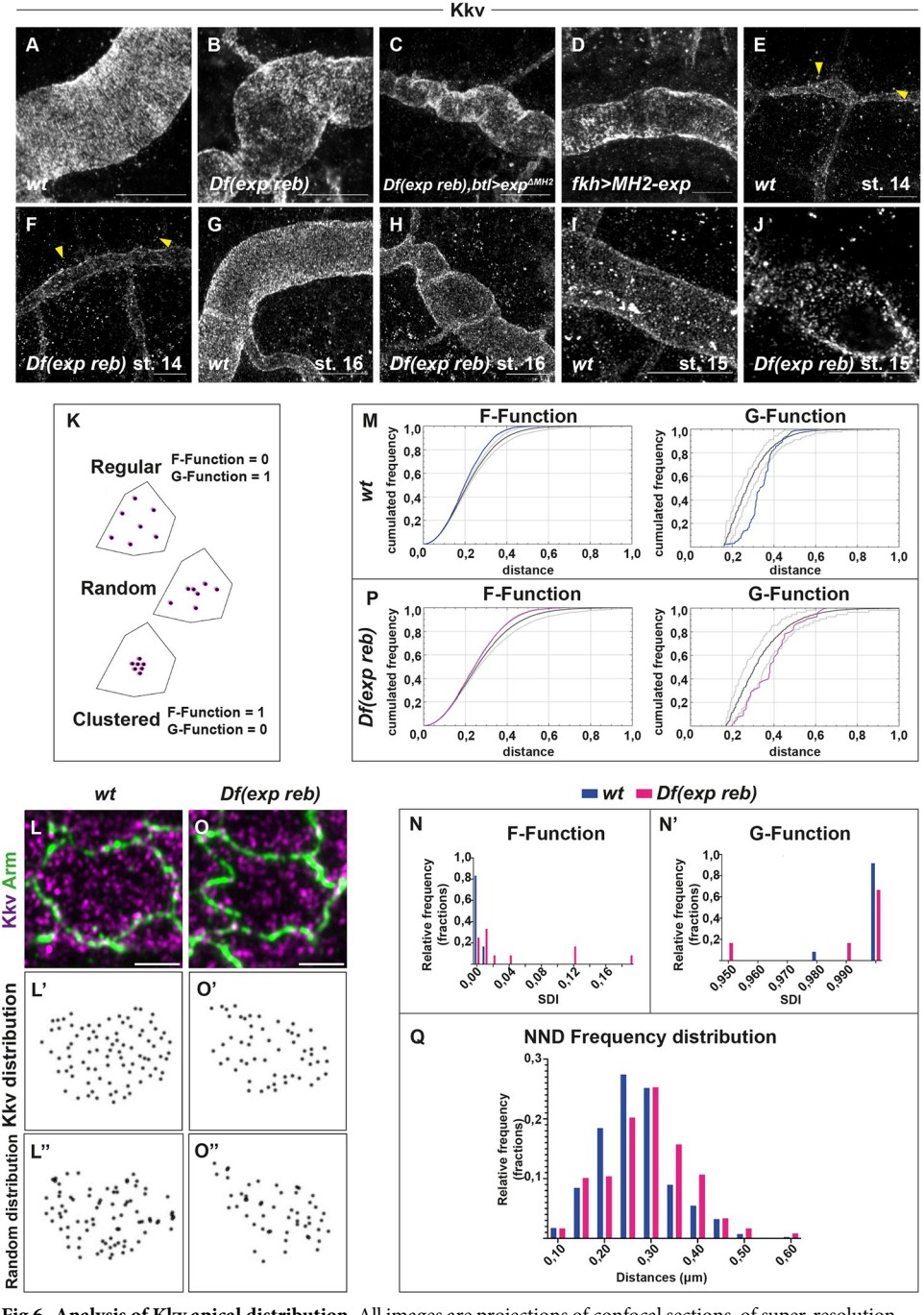

**Fig 6. Analysis of Kkv apical distribution.** All images are projections of confocal sections, of super-resolution microscopy. (**A**, **B**) Kkv localises apically in the trachea of wild-type embryos (**A**) and in absence of *exp reb* (**B**). (**C**, **D**) The localisation of Kkv is apical also in presence of *exp* $^{\Delta MH2}$ in trachea (**C**) and in presence of *MH2-exp* in salivary glands (**D**). (**E**, **F**) At stage 14, in wild-type embryos (**E**) and in embryos deficient for *exp* and *reb* (**F**), Kkv is present in the apical membrane and in many intracellular vesicles (yellow arrowheads). (**G**) At stage 16, in wild-type embryo, Kkv apical distribution follows the pattern of taenidial folds and intracellular vesicles are mostly absent. (**H**) At stage 16, in *exp reb* mutant embryos, Kkv is apical but shows altered distribution pattern. (**I**, **J**) At stage 15, in control embryos, Kkv pattern is apical and covers the whole membrane leaving minimal spatial gaps (**I**); instead, in *exp reb* mutant embryos, Kkv distribution changes to a less organised pattern at the apical membrane (**J**). (**K**) Three different types of spatial distribution within a selected area. The positions of the defined objects can be random and exhibit characteristics of attraction (clustered pattern) or repulsion (regular pattern). The F-Function tends to be larger (≈1) for clustered patterns and smaller (≈0) for regular. The G-Function tends to be smaller (≈0) for clustered and larger (≈1) for regular patterns. (**L**) Kkv punctae (magenta) on the apical cell area marked by Armadillo (green) in the

trachea of a control embryo. (**L'**) Positions of Kkv punctae on the selected area marked by black dots. (**L"**) Random pattern of distribution for the same area created by the spatial statistics 2D/3D image analysis plugin. (**M**) The corresponding observed F and G functions (blue) are displayed above and below the reference simulated random distributions (black) and the 95% confidence interval (light gray), respectively, indicating a nonrandom spatial pattern. (**N**) SDI histogram for the F-Function of the control (blue) and the *Df(exp reb)* samples. A significant difference between the frequency distributions for each group of individuals has been observed. (Kolmogorov–Smirnov D = 0.5833, $p < 0.05$) (**N'**) SDI histogram for the G-Function of the control (blue) and the *Df(exp reb)* samples. Statistical analysis of the distributions did not reveal significant differences between the two groups of individuals for this parameter (Kolmogorov–Smirnov D = 0.25, $p > 0.05$). (**O**) Kkv punctae (magenta) on the apical cell area marked by Armadillo (green) in the trachea of a *exp reb* mutant embryo. (**O'**) Positions of Kkv punctae on the selected area marked by black dots. (**O"**) Random pattern of distribution for the same area created by the spatial statistics 2D/3D image analysis plugin. (**P**) The corresponding observed F and G functions (blue) are displayed above and below the reference simulated random distributions (black), respectively. Both curves largely overlap with the 95% confidence interval (light gray), indicating a tendency towards a random spatial pattern. (**Q**) Frequency distribution histograms for the nearest neighbour distances between Kkv punctae in control (blue) and *exp reb* mutant samples. The distribution of values between the two groups is found significantly different (Kolmogorov–Smirnov D = 0.2036, $p < 0.005$). The underlying data for quantifications can be found in the S1 Data. Scale bars **A-J**: 10 μm; **L**, **O**: 2 μm.

calculated (Fig 6K), followed by the spatial distribution index (SDI), which accounts for the difference between the observed pattern and a completely random one [26–28]. For randomly organised patterns, the SDI within a population should be uniformly distributed between 0 and 1. Deviations from spatial randomness are expected to shift the SDI values closer to 0 or 1, indicating regular (F–SDI = 0, G–SDI = 1) or clustered (F–SDI = 1, G–SDI = 0) organisation. The results of this analysis showed that Kkv's localisation on the apical membrane appears to have an underlying spatial order (Fig 6L, 6L', 6L" and 6M). The SDI for the F-Function values for the control samples analysed were distributed between 0 and 0.01, while the one for the G-Function values were distributed between 0.98 and 1 ($n = 12$) (Fig 6N and 6N'). The same analysis for the *exp reb* mutants (Fig 6O, 6O', 6O" and 6P) revealed low F and high G-function SDI values, within a range of 0 to 0.19 and 0.95 to 1 ($n = 12$), respectively (Fig 6N and 6N'). Although this could also resemble a nonrandom pattern within a population, in our comparisons, the distribution of the F-Function SDI values of the controls was significantly different from the *exp reb* mutants (Kolmogorov–Smirnov D = 0.5833, $p < 0.05$) (Fig 6N), indicating a shift of the Kkv organisation towards "randomness." Comparing the G-Function SDI value distributions (Fig 6N') could not reveal significant differences between the control and the *exp reb* mutants (Kolmogorov–Smirnov D = 0.25, $p > 0.05$); however, this was not surprising as this parameter is less sensitive to change and thus to detect variation among samples [26]. For this reason, we directly calculated the nearest neighbour distances (NNDs) of the Kkv puncta from all control and *exp reb* mutant samples, and we created the frequency distribution plots for the values obtained (Fig 6Q). By this approach, we were able to detect a significant shift of the distribution frequencies between the two groups (Kolmogorov–Smirnov D = 0.2036, $p < 0.005$), further supporting the hypothesis of changes in Kkv apical distribution upon removal of *exp* and *reb*.

To examine whether the spatial pattern of Kkv is directly affected by modifications of Exp and Reb levels and is not due to possible perturbations derived from lack of extracellular chitin in the trachea in *exp/reb* mutants, we ectopically expressed *reb* in salivary glands. As indicated previously, salivary glands normally express *kkv* (S4A Fig) but lack *reb* expression. For this reason, we examined the localisation of Kkv, analysing the F-Function values and the NND for the detected punctae, in cells of the salivary glands in control embryos and in embryos expressing *reb* (*fkh-Gal4>UAS-reb*) (S5A–S5F Fig) ($n = 12$ in each case). We observed that also in this tissue, the distribution of Kkv seems to be affected by the ectopic expression of *reb*. The distribution of the F-Function SDI values of the controls (calculated between 0 and 0.09) was significantly different from the *reb* expressing embryos (calculated between 0 and 1) (Kolmogorov–

Smirnov D = 0.6667, $p < 0.05$) (S5G Fig). Comparing the distribution frequencies of the NND between the two groups also revealed a significant shift (Kolmogorov–Smirnov D = 0.1463, $p < 0.05$) (S5H Fig). Interestingly, the analysis of control salivary glands where *exp* and *reb* are absent showed a tendency of Kkv distribution towards spatial randomness, similar to the observation in the trachea of *exp reb* mutants.

All these results suggested that *exp/reb* are directly or indirectly required for the proper distribution of Kkv at the apical membrane.

## 7. Exp/Reb and Kkv localise in a complementary pattern

As Exp/Reb proteins localise at the apical membrane [13] and we found that Exp/Reb are required for Kkv apical distribution, we investigated their relative localisations using super-resolution microscopy. Analysis of endogeneous Reb and Kkv proteins in the trachea of wild-type embryos indicated that the two proteins in general do not colocalise. Rather, it seemed that they showed a complementary pattern, where Reb accumulated between the Kkv accumulation (Fig 7A). We looked at the pattern of Reb in relation to Kkv at stage 16, when Kkv accumulates in the taenidia, to investigate whether Reb is involved in generating this pattern of Kkv. We observed that Reb showed a complementary pattern to that of Kkv at a fine scale rather than accumulating in complementary stripes to those of Kkv (Fig 7B). This result suggested that Exp/Reb may regulate Kkv distribution at the local subcellular level. To confirm the complementary pattern of Kkv and Reb observed in the trachea, we looked at salivary glands expressing Reb. We also found that the Kkv and Reb pattern were complementary and that Reb accumulated particularly where Kkv is low (Fig 7C).

To further investigate whether Kkv and Exp/Reb interact, we performed coimmunoprecipitation experiments. Our αKkv and αReb antibodies recognised Kkv and Reb in embryo extracts (S5 Fig). While αKkv antibodies immunoprecipitated the protein with high efficiency (S5J Fig), they did not coimmunoprecipitate Reb (S5K Fig). This negative result may be due to technical difficulties and does not fully rule out an interaction between Kkv and Reb. However, the biochemical and super-resolution analyses do not support a physical interaction between Kkv and Reb.

In summary, our analyses of Exp/Reb and Kkv accumulation suggested that *exp/reb* are indirectly required for the proper distribution of Kkv at the apical membrane.

## Discussion

### Dissection of the roles of conserved motifs in Exp/Reb and Kkv proteins

We generated different transgenic lines under the control of UAS to investigate the function of different conserved domains in Exp/Reb and Kkv proteins. The Gal4/UAS experimental approach allowed us to test whether the mutated proteins are sufficient to promote ectopic or advance chitin deposition (as wild-type proteins do), and also whether the mutated proteins can rescue chitin deposition in a mutant background for *exp/reb* or *kkv*, respectively [29]. However, the Gal4/UAS approach has also different caveats, as it involves the overexpression of the protein, which can bypass or mask the endogenous requirements of specific protein domains, allowing reduced activity mutations to become active.

Our structure–function analysis identified a critical role for the Nα-MH2 domain of Exp/Reb in extracellular chitin translocation. However, the domain was dispensable for chitin polymerisation and for Exp/Reb and Kkv localisation at the apical membrane. The Nα-MH2 domain alone was not able to localise to the membrane and was not functional either. We suggest that Exp/Reb need to localise apically to be active and to promote extracellular chitin translocation. In this context, the Nα-MH2 would be required for interaction/recruitment of

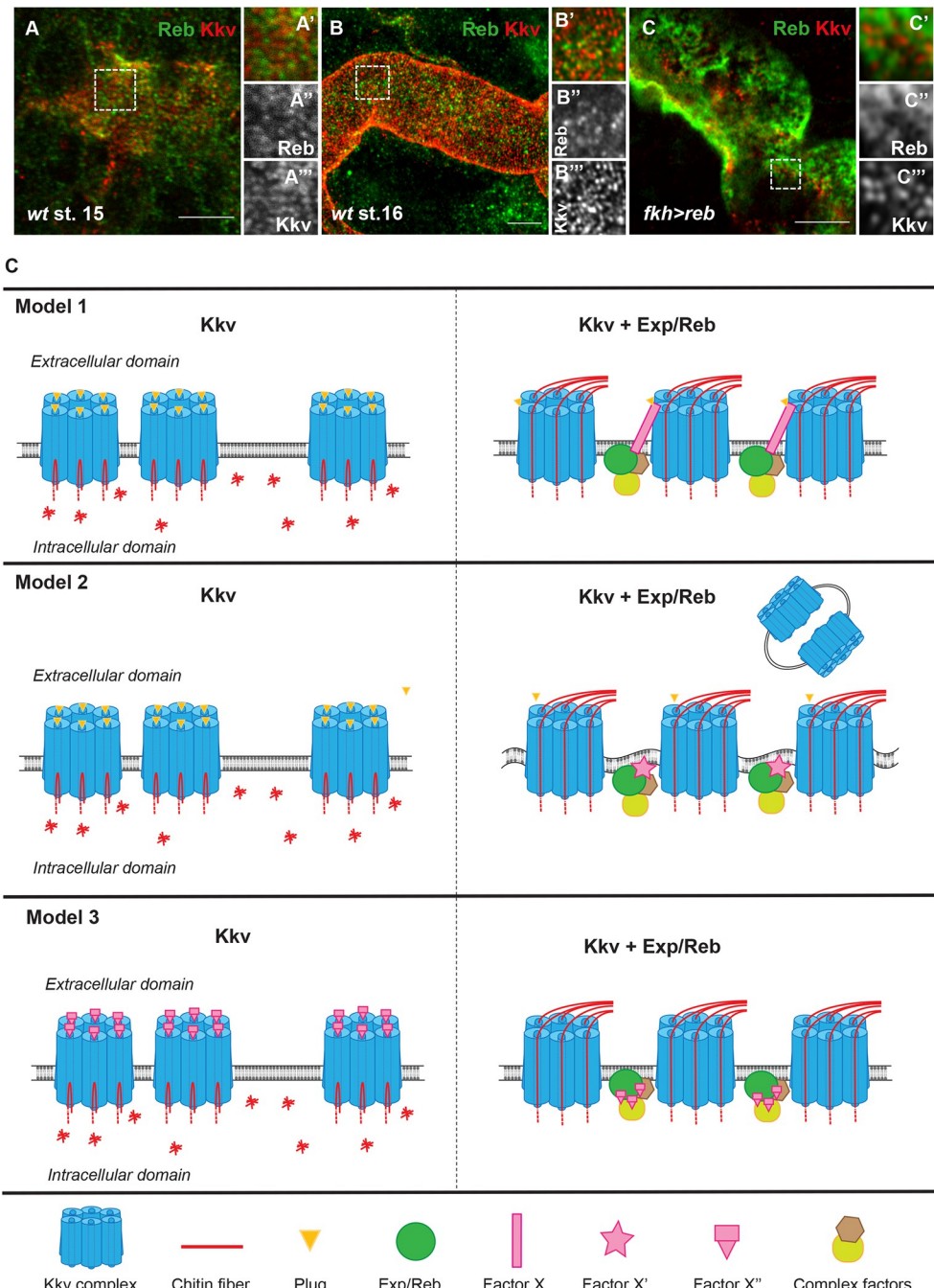

**Fig 7. Accumulation of Kkv and Reb.** All images are super-resolution single confocal sections except B, which is a projection of super-resolution confocal sections. (**A**, **B**) In the trachea of wild-type embryos, Reb and Kkv do not colocalise, and they show a complementary pattern (**A'-A'''**) at the local subcellular level. (**C**) In salivary gland of embryos expressing Reb, the patterns of Kkv and Reb are complementary. (**D**) Models for the role of *kkv* and *exp/reb* in chitin deposition. Kkv oligomerises in complexes that localise to the apical membrane (as proposed in [2]). In the absence of *exp/reb* activity, Kkv can polymerise chitin from sugar monomers (discontinuous red lines), but it cannot translocate it because the channel is closed, and polymerised chitin remains in the cytoplasm. In addition, Kkv is not homogeneously distributed. Exp/Reb form a complex with other proteins, which localises to the apical membrane. The presence of Exp/Reb complex regulates Kkv apical distribution and activity. In model 1, we propose that a factor/s recruited by Exp/Reb (Factor X) can induce a posttranslation or conformational modification to Kkv protein that opens the channel promoting translocation of chitin fibers to the extracellular domain. In model 2, we propose that a factor/s recruited by Exp/Reb (Factor X') can induce changes in membrane composition/curvature that will then promote a conformational change in Kkv that opens the channel to translocate chitin. These membrane changes lead

to Kkv shedding extracellularly. In model 3, we propose that Exp/Reb complex can bind and relocalise Factor X", which normally inhibits Kkv-translocating activity. This neutralises the activity of Factor X" allowing chitin translocation. Scale bars: 5 μm.

other protein/s, presumably in a complex at the apical membrane, which would confer to Kkv the ability to translocate the synthesised polymers. In line with this hypothesis, we identified a new highly conserved motif with a role in Exp/Reb apical accumulation. In our experimental conditions, the CM2 motif was dispensable for *exp/reb* function in extracellular chitin deposition; however, we found that in spite of a clear effect in protein localisation, Exp$^{\Delta CM2}$ protein was still able to localise apically, which could account for its activity. We speculate that the massive amount of Exp$^{\Delta CM2}$ protein produced in our experimental Gal4/UAS conditions is masking the requirement of the domain for the proper localisation. The generation of CRISPR mutants lacking the CM2 will help to validate this hypothesis in future work.

Kkv is a huge protein with several recognisable domains [11,15]. Our analysis of the conserved WGTRE domain of Kkv indicated that it is required for ER exit. Kkv is produced at the ER and traffics to the membrane where it localises apically to deposit chitin extracellularly. In the case of the yeast CHS Chs3, ER exit requires the activity of the DHHL protein Pfa4 and the dedicated chaperone Chs7 [30,31]. Hence, ER exit of CHS seems to be a strictly regulated step. In line with this, recent work identified a sarco/endoplasmic reticulum $Ca^{2+}$-ATPase, Serca, which physically interacts with Kkv and regulates chitin deposition [32]. The WGTRE motif could directly bind Serca and/or other proteins for ER exit.

The CC domain of Kkv is predicted to face the extracellular space and may be involved in protein–protein interactions or oligomerisation [8,16]. Our results indicated that Kkv$^{\Delta CC}$ behaves as the full-length Kkv in several conditions, suggesting that the CC domain is dispensable for Kkv activity. However, we also found that while the coexpression of *reb* and full-length *kkv* in the trachea leads to abnormal tube morphogenesis (due to excessive chitin), the concomitant overexpression of *reb* and *kkv$^{\Delta CC}$* resulted in normal tube formation. This is an intriguing behaviour, because the sole overexpression of *reb* in an otherwise wild-type condition (i.e., in presence of endogenous *kkv*) produces an abnormal tube morphogenesis. Therefore, the result suggested that *kkv$^{\Delta CC}$* is acting to decrease *kkv* endogenous activity. This result would fit with the hypothesis that the CC domain is involved in the oligomerisation of Kkv units. In our model, Kkv$^{\Delta CC}$ pure oligomers and Kkv$^{WT}$ oligomers would be fully functional, whereas Kkv$^{\Delta CC}$-Kkv$^{WT}$ would not be, maybe due to conformational incompatibilities. In a null mutant background for *kkv*, only Kkv$^{\Delta CC}$ oligomers would be present and would support extracellular chitin deposition. In contrast, in a wild-type background, Kkv$^{\Delta CC}$-Kkv$^{WT}$ inactive oligomers would form, limiting the capacity to promote increased chitin deposition. In support of this hypothesis, it has been proposed that CHS may work as oligomers, forming complexes to produce chitin [15,33–35]. On the other hand, we also find that the absence of the CC domain renders a protein that is less efficient to localise apically and that displays a different turnover compared to the wild-type Kkv. We propose that this protein would be less active than the wild-type form, but, again, its overexpression would mask this decreased activity. In any case, our results indicate that the CC is involved in protein localisation and trafficking and point to a correlation between Kkv activity and Kkv trafficking and localisation (see below).

## Kkv and chitin trafficking

Salivary glands proved to be an excellent in vivo test tube to investigate Kkv biology on its own and in relation to chitin deposition. Salivary glands do not deposit chitin in normal conditions. Overexpression of *kkv* (in absence of *exp/reb* activity) promotes polymerisation of chitin that

accumulates intracellularly. The simultaneous expression of *kkv* and *exp/reb* leads to ectopic extracellular chitin deposition in the lumen.

Using a GFP-tagged Kkv protein, we followed Kkv trafficking and localisation. We found that Kkv traffics via Golgi, reaches the apical membrane, is then internalised, and finally degraded or recycled back to the membrane. A comparable trafficking route has been described for CES, CSC [36]. Also, in line with results with CSC, our results indicated a correlation between Kkv activity and Kkv trafficking and localisation [36]. Kkv protein unable to polymerise chitin did not localise apically and was found diffused in the cytoplasm, strongly suggesting that Kkv polymerisation is important for Kkv localisation/stabilisation at the membrane, as also proposed by other labs [16]. On the other hand, fully active Kkv protein able to polymerise and translocate chitin extracellularly becomes strongly enriched apically. We detected some Kkv recycling under these conditions, but, strikingly, we also detected the presence of Kkv punctae in the luminal extracellular space. These extracellular Kkv punctae, which correspond to membranous vesicles, suggested that the protein is shed during or after chitin deposition. Kkv shedding was previously documented in sensory bristles [16,23]. Because the GFP tags that allowed us and others [16] to visualise Kkv shedding is cytoplasmic, the results indicate that the whole Kkv protein is shed, rather than cleaved. Membrane proteins can be shed to the extracellular space through exosomes or microvesicles [24,25]. Exosomes derive from endosomal trafficking, but we find that Kkv extracellular vesicles still form when we block endocytosis. Thus, we propose that Kkv is shed through microvesicles when it is actively extruding chitin. Microvesicles arise by outward blebbing and pinching off the plasma membrane, releasing the membrane protein to the extracellular space. Microvesicle formation requires redistribution of membrane lipid and protein components, which modulate changes in membrane curvature and rigidity [24,25]. This suggests that conferring the translocating ability to Kkv by *exp/reb* activity is linked to membrane reorganisation.

Overexpression of *kkv* in the absence of *exp/reb* activity showed that Kkv is able to polymerise chitin throughout all its trafficking route as we detected exocytic, endocytic, and recycling Kkv vesicles partially colocalising with chitin. This indicates that Kkv is already assembled as a functional synthesising complex in Golgi (but not in ER, as in ER retention conditions, no chitin polymerisation is detected), before reaching the membrane, and once it is internalised, as shown for the CSC [36]. Intriguingly, we also detected the presence of large amounts of membrane-less aggregates of chitin, free of Kkv, in the cytoplasm under these conditions. The exact origin of these chitin aggregates is unclear; however, our observations and previously published work lead us to propose the following model. It has been reported that Kkv localises to the membrane with the catalytic domain facing the cytoplasm [8,16]. Therefore, Kkv would also localise with its catalytic domain facing the cytoplasm in intracellular vesicles (either exocytic, endocytic, or recycling). Kkv would polymerise chitin in a constitutive manner once it is assembled in Golgi; however, it would not be able to translocate chitin unless *exp/reb* activity is present, that is in the apical membrane. In the absence of *exp/reb*, the chitin polymerised by Kkv during trafficking or from the apical membrane would not be translocated and would be instead abnormally released intracellularly. The released fibers would remain in the cytoplasm and could organise in aggregates by phase separation [37]. Actually, chitin fibrils are insoluble at physiological PH [5]. In agreement with this model, we find huge amounts of chitin at the subcortical level (where large amounts of Kkv protein localise). We also find that in common punctae, Kkv and chitin normally show partial colocalisation, or they appear in close contact, which would reflect the process of polymerisation of chitin facing the cytoplasm. Our live-imaging analysis supports this model as we observed partial colocalisation of Kkv and chitin (corresponding to chitin polymerisation) and the separation of punctae (corresponding to the

release of chitin fibers). While polymerising chitin constitutively may be a general mechanism of activity of Kkv, that we detect in conditions of Kkv overexpression, the fact that we do not detect chitin intracellularly produced by endogenous levels of Kkv suggest that this may be a residual effect of Kkv activity in wild-type conditions.

## Role of Exp/Reb in chitin deposition

It has been proposed that chitin polymerisation occurs in the cytoplasm, where the catalytic domain of CHS localises [8,9,16]. Our results confirm that chitin is polymerised intracellularly. It has also been proposed that chitin polymerisation is tightly coupled to chitin extrusion [2]. In this respect, it has been proposed that the structural organisation of the transmembrane helices of CHS would form a pore or central channel through which the nascent polymer would be translocated [1,2,8,9,38]. Our results, however, indicate that, at least for the case of the CHS Kkv, the polymerisation and extracellular translocation of chitin fibrils are uncoupled and that while Kkv is sufficient to polymerise the polysaccharide, it cannot translocate it without the activity of *exp/reb*. Thus, we identify a role for Exp/Reb in promoting Kkv-mediated chitin translocation. We cannot exclude that, besides this role, Exp/Reb perform other roles, acting, for instance, as processivity factors regulating the elongation of chitin polymers, as in their absence Kkv synthesise short polymers that remain in the cytoplasm. Further work will be required to identify additional roles for Exp/Reb.

What is then the role of *exp/reb* in chitin deposition? Exp/Reb could potentially play a role in extracellular chitin deposition by binding nascent fibrils polymerised by Kkv and assisting their proper translocation. However, it is unlikely that Exp/Reb could play this role as (1) no chitin binding domain and (2) no transmembrane domain that could form a pore or channel have been identified in these proteins. On the other hand, we do not support either a model in which Exp/Reb form a complex with Kkv, based on the following observations: (1) super-resolution analysis revealed a complementary pattern of Exp/Reb and Kkv at the apical membrane; (2) we could not detect a physical interaction between Reb and Kkv in coimunoprecipitation experiments; and (3) recent work searching for Kkv interacting proteins did not identify Exp or Reb [32,39].

Our results indicate that *exp/reb* activity is required for the proper distribution and clustering of Kkv at the apical membrane. It has been proposed that CHS are organised at the plasma membrane in a quaternary structure forming rosettes [2]. This hypothesis comes from the comparison between CHS and the closely related CES: Both enzymes belong to the β-glycosyl-tranferase family, produce polymers with similar molecular structure, and share several conserved motifs like the catalytic domain QxxRW and several transmembrane domains [8,40]. CES organise as hexagonal structures with 6-fold symmetries (rosettes). Each rosette is composed of six subunits/lobes, which, in turn, consist of either six monomeric or three dimeric synthetic units, each capable of polymerising and translocating a single cellulose chain. It has been proposed that differences in the morphology of either the rosettes or their lobes may be responsible for the diversity in cellulose architecture, for instance, dispersed rosettes produce widely spaced cellulose microfibrils, whereas dense regions of complexes synthesize highly aggregated crystalline microfibrils (for reviews, see [36,41]). CHS, like Kkv, may function in a similar way [2]. In this scenario, we propose that exp/reb controls the distribution/clustering of Kkv rosette complexes, possibly regulating the spacing by localising in a complementary pattern, which would, in turn, regulate Kkv activity.

We envision at least three different models, or a combination of them, for a role of *exp/reb* in chitin translocation (Fig 7D). In the different models, Kkv has a closed conformation in the absence of Exp/Reb that prevents chitin translocation, and Exp/Reb regulate the apical

distribution of Kkv. In our first model, Exp/Reb are required to organise a complex (through its Nα-MH2 domain) that reaches the apical membrane (through its CM2). One of the components of the complex, Factor X, interacts directly with Kkv promoting a posttranscriptional modification or a conformational change in Kkv that activates its translocation activity. In the absence of Exp/Reb, Factor X would not reach the membrane and Kkv would remain plugged or clogged preventing the translocation of the chitin fibrils. In a second model, Exp/Reb are required (in collaboration with its complex) to promote a favourable membrane environment (for instance, curvature, membrane composition) for the proper integration/insertion/arrangement of Kkv rosette complexes. This arrangement of the complex is required to impose a particular conformational organisation in the rosette that opens the pore of the Kkv units, which otherwise are plugged or clogged. In agreement with this second model, it has been suggested that the Kkv-interacting protein Ctl2 could regulate membrane phospholipid composition and increase Kkv activity [39]. In addition, this model would fit with the putative role of membrane curvature and composition and Kkv shedding as microvesicles [24,25]. A similar mechanism has been shown for the mechanosensor Piezo, where the conducting conformation of its pore (open or closed) can be controlled by local changes in membrane curvature and tension [42,43]. In a third model, in the absence of Exp/Reb complex, Kkv translocating activity is repressed by interactions between Kkv and a Kkv inhibiting factor (Factor X"). Exp/Reb, or another member of the complex, can bind Factor X" releasing its interaction with Kkv, thereby neutralising its activity and allowing Kkv-mediated chitin translocation.

Remarkably, a very recent paper published during the revision of this manuscript identified a "gate lock" in the chitin-translocating channel of *Phytophthora sojae* Chs1, which ensures the growing of the oligomer and directs the polymer through the exit of the channel towards the extracellular domain [44]. The domain serving as the "gate lock" is highly conserved among CHS, and the relevant residues of the domain are present in Kkv [44]. Altogether, these recent structural data reinforce the model we propose and suggest a central role for Exp/Reb in regulating this "gate lock."

In summary, our results point to an absolute requirement for *exp/reb* in regulating the capacity of Kkv to translocate the nascent chitin fibrils, likely regulating its distribution and conformational organisation at the apical membrane. This reveals the existence of an extrinsic mechanism of regulation of CHS, which seems to be conserved during evolution, as orthologs for Exp have been identified in arthropods and nematodes.

## Materials and methods

### *Drosophila* strains and maintenance

All *Drosophila* strains were raised at 25˚C under standard conditions. Balancer chromosomes were used to follow the mutations and constructs of interest in the different chromosomes. For overexpression experiments, we used the Gal4 drivers *btlGal4* (in all tracheal cells, kindly provided by S. Hayashi) and *fkhGal4* (in salivary glands, kindly provided by D. Andrew). The overexpression and rescue experiments were performed using the Gal4/UAS system [29]. To maximise the expression of the transgenes, crosses were kept at 29˚C.

The following fly strains were used: *kkv^IB22^*, *kkv^63−20^*, *UAS-GFPKkv* (kindly provided by B. Moussian), *UAS-verm UAS-serp* (kindly provided by S. Luschnig). The following stocks were obtained from Bloomington Drosophila Stock Center (BDSC): y$^1$w$^{1118}$ (BDSC#6598, used as wild type), *Df(2R)BSC879* (BDSC#30584, deficiency for *exp* and *reb*), *UAS-reb* (*reb^LA0073^*, BDSC#22192), *UAS-ChtVis-Tomato (*BDSC#66512, to visualise chitin live), *UAS-CD4mIFP (BDSC#64183)*. The following stocks were generated in our lab: *UAS-Exp^ΔMH2^*, *UAS-Reb ^ΔMH2^*, *UAS-MH2-exp*, *UAS-MH2-reb*, *UAS-Exp^ΔCM2^*, *UAS-GFPKkv^ΔWGTRE^*, *UAS-GFPKkv^ΔCC^*, *btl-Gal4-UAS-srcGFP* (recombinant line to visualise tracheal cells).

## Immunohistochemistry

Embryos were stained following standard protocols. Embryos were staged as described [45]. Embryos were fixed in 4% formaldehyde (Sigma-Aldrich) in PBS1x-Heptane (1:1) for 10 min for Ecad staining and for 20 min for the rest. Embryos transferred to new tubes were washed in PBT-BSA blocking solution and shaken in a rotator device at room temperature. Embryos were incubated with the primary antibodies in PBT-BSA overnight at 4˚C. Secondary antibodies diluted in PBT-BSA (and for the CBP staining) were added after washing and were incubated at room temperature for 2 to 5 h in the dark. Embryos were washed, mounted on microscope glass slides, and covered with thin glass slides.

The following primary antibodies were used: rat anti-Exp (1:100), rabbit anti-Reb (1:100), and rat anti-Reb (1:100 for IF; 1:6,000 for WB) [13]; rabbit anti-Kkv (1:100 for IF; 1:4,000 for WB, this work), rabbit anti-Cp190; rabbit anti-Rab11 (1:2,000, kindly provided by T. Tanaka); rabbit anti-Serp (1:200) and rabbit anti-Verm (1:200), kindly provided by S. Luschnig); rabbit anti-Perlecan (1:200, kindly provided by A. González-Reyes); mouse anti-FK2 (1:50, Enzo Life Science); goat anti-GFP (1:600, AbCam); rabbit anti-GFP (1:600, Thermo Fisher Scientific–Invitrogen); mouse anti-KDEL (1:200, Stressmarq Biosciences); rabbit anti-Arl8 (1:100, DSHB#2618258), mouse anti-Armadillo (1:100, DSHB#528089), rat anti-E-Cadh, DCAD2 (1:100, DSHB#528120), goat anti-Golgin245 (1:2000, DSHB#2569587), mouse anti-Hrs-27-4 (1:10, DSHB#2618261), mouse anti-α-Spec (1:10 DSHB#528473), mouse anti-Gasp 2A12 (1:5 DSHB#528492) from Developmental Studies Hybridoma bank-DSHB.

Cy3-, Cy2-, and Cy5-conjugated secondary antibodies (Jackson ImmunoResearch) were used at 1:300. Chitin binding probe fluorescently labelled CBP (1:300) was used to visualise chitin (kindly provided by N. Martín, from J. Casanova lab).

## Image acquisition

Fluorescence confocal images of fixed embryos were obtained with Leica TCS-SPE system using 20× and 63× (1.40–0.60 oil) objectives (Leica). For super-resolution images, two different systems were used: Elyra PS1- Airyscan (Zeiss) from the IRB-Advanced Digital Microscopy Core Facilty and Drangofly 505 (Andor) from IBMB-Molecular Imaging Platform; in both cases, a 100× (1.40–0.60 oil) objective was used. The latter system was used also to perform life imaging. In this case, dechorionated embryos were mounted and lined up on a Menzel-Glaser coverslip with oil 10-S Voltalef (VWR) and covered with a membrane (YSI membrane kit). In all movies, we used 63× (1.40–0.60 oil) objective. To visualise time-lapse movies, single sections were used. Fiji (ImageJ) [46] was used for measurement and adjustment. Otherwise indicated in the text, confocal images are maximum-intensity projections of Z-stack sections. Figures were assembled with Adobe Illustrator.

## Generation of UAS constructs

For the generation of new recombinant DNA, we used pUAST-exp, -reb, and -GFPkkv [13]. We digested the DNA from the vector using the following couples of restriction enzymes (New England BioLabs, NEB): EcoRI/XhoI for *exp*, EcoRI/NotI for *reb*, and XhoI/XbaI for *GFP-kkv*, and we cloned them in the vector pJET1.2.

The constructs UAS-exp$^{\Delta MH2}$, -reb$^{\Delta MH2}$, -exp$^{\Delta CM2}$, -GFP-kkv$^{\Delta WGTRE}$, and -GFP-kkv$^{\Delta CC}$ were obtained by directed deletion. To delete specific regions from the DNA, "Q5 Site-Directed Mutagenesis Kit" (NEB) was used. The kit comprehended material to perform PCR, ligation of the fragments, and transformation of NEB-α competent cells. Deletions were created by designing primers that flank the region to be deleted, then we performed a PCR obtaining a linear double filament of DNA composed by the original pJET1.2 vector and the flanking

regions of the deleted sequence. Upon ligation of the fragment, competent cells were transformed and plated in selective plates. Miniprep to obtain DNA were performed using the kit NZYtech, and the DNA was sequenced through the platform Eurofins Genomics. Finally, the new mutated DNA was digest using the restriction enzymes described above and cloned in pUAST vector for exp$^{\Delta MH2}$ and reb$^{\Delta MH2}$ and in pUAST-attB vector for all the other DNAs. After performing a midiprep (NZYtech), the DNAs were injected in embryos y$^1$w$^{1118}$ by the "Drosophila injection Service" of the "Institute for Research in Biomedicine" (IRB, Barcelona) and by the "Transgenesis Service" of the "Centro de Biología Molecular Severo Ochoa" (CBM, Madrid).

UAS-MH2-exp and UAS-MH2-reb were obtained amplifying the MH2 region, respectively, from pJET-exp and pJET-reb and cloning the fragment in the pUAST-attB vector using in both cases EcoRI/NotI.

The primers used in this study are the following:

PCR primers to delete the Nα-MH2 domain in *expansion*: sense 5′-GTG GTG GCC ATG GAT ATG-3′ and antisense 5′-GTC GAT TTG GGT CCA TTTG-3′;

PCR primers to delete the Nα-MH2 domain in *rebuf*: sense 5′-AAC GTG GTG GCC ATG GAC-3′ and antisense 5′-GTC GAT TTG CTC CCA CTTG-3′;

PCR primers to delete the CM2-8 domain in *expansion*: sense 5′-CGG GCC CGA GTT CCG AAC-3′ and antisense 5′-ATA TTT CTT ATT ATC TTT GCC CTT GTC AGA TTT ACC-3′;

PCR primers to delete the CM2-17 (long) domain in *expansion*: sense 5′-TCC GGC AAG CCG ATA CCC-3′ and antisense 5′-ATA TTT CTT ATT ATC TTT GCC CTT GTC AGA TTT ACC-3′;

PCR primers to delete the WGTRE domain in *GFPKkv*: sense 5′-GTG GTG GCT AAG AAG ACC-3′ and antisense 5′-GGA GAC GAC GTT TAG GTTG-3′;

PCR primers to delete the CC domain in *GFPKkv*: sense 5′-AGC ATG CTG AGC TTC CTTC-3′ and antisense 5′-GGT CTT CTT AGC CAC CAC-3′;

PCR primers to clone MH2-exp: sense 5′-CGG AAT TCA TGG ACG AGA TCT GGG CCAA-3′ and antisense 5′-ATA GTT TAG CGG CCG CTC ACG GGC GGT TCT TGA -3′;

PCR primers to clone MH2-reb: sense 5′-CGG AAT TCA TGG ACG AGA TCT GGG CCAA-3′ and antisense 5′-ATA GTT TAG CGG CCG CTC AGC TGC TGT TCG TCAG-3′;

## Generation of antibodies

To generate polyclonal antibody against Kkv, fragments were amplified by PCR using the following primer combination: sense 5′-GGA ATT CCA TAT GGG AAT CGA TGG CGA CTAC-3′ and antisense 5′-CCG CTC GAG TCA CAG GCG ACC TGT GCC ATT-3′. We used the restriction sites NdeI/XhoI. The amplified fragments were cloned into the expression vector pET14b (Novagen). The resulting positive clones were used to transform BL2 (C41) cells (Novagen) for protein expression. Cells were induced with 1 mM IPTG, and proteins were expressed at 37°C during 2 h. The positive clones were selected, and the recombinant proteins (22 KDa) fused with a His tag were purified through a column of nickel (Quiagen) in denaturalising conditions (8 M urea). The purified proteins were used to inject rabbits by the facility CID-CSIC-Production of antibodies (Barcelona).

## Quantification and statistical analysis

Data from quantifications were imported and treated in the Excel software and/or in Graph-Pad Prism 9.0.0, where graphics were finally generated. Graphics shown are scatter dot plots, where bars indicate the mean and the standard deviation. Statistical analyses comparing the

mean of two groups of quantitative continuous data were performed in GraphPad Prism 9.0.0 using unpaired two-tailed Student *t* test applying Welch's correction. Differences were considered significant when $p < 0.05$. Significant differences are shown in the graphics as $^{*}p < 0.05$, $^{**}p < 0.01$, $^{***}p < 0.001$, $^{****}p < 0.0001$. n.s. means not statistically significant. Sample size (n) is provided in the figures or legends. The nonparametric Kolmogorov–Smirnov test was used for the comparisons of frequency distributions.

### Image analyses

**Apical/Basal localisation.**   On single section images, we analysed the intensity of the protein localised at the apical and basal membranes of single cells. The apical domain was identified by staining with E-cadherin or Armadillo, the basal domain with the general membrane markers α-Spectrin or Src-GFP or with the basement membrane marker Perlecan. Line-shaped ROIs of 15 pixels width were drawn with the "free hand line" tool to select the apical and the basal domain of the same cell (respectively, orange line and yellow line in Figs 2E and 3K). The integrated density (IntDen) of the protein for each ROI was quantified using the measure tool of Fiji software. The values obtained can be found in the S1 Data. Each value of the graphs represents the ratio between the IntDen of the apical region and the IntDen of its respective basal region. The n represents the total number of ratios calculated and, in brackets, the number of embryos analysed.

### Number of vesicles/particles

Maximum intensity projections of 13 sections (0.29 μm each) and Fiji software were used. After subtracting the background, an ROI was drawn around the salivary gland excluding the apical membrane (S2E Fig). A binary mask was created using the threshold tool and the watershed segmentation tool (S2E' Fig). Numbers of vesicles were counted using the Analyse Particles tool, and the parameters were set to 0.02 to 1.1 μm² size, 0 to 1 circularity; the number of vesicles and a mask of the result was obtained. The values obtained can be found in the S1 Data.

### Colocalisation of vesicles/particles

The "And" operation of the Image Calculator tool of Fiji software was applied between masks obtained as a result of the "Number of vesicles" process. The resulting image was analysed through the Analyse Particle tool as described above.

### Analysis of Kkv distribution

For the Kkv distribution analysis in tracheal cells, we used maximum intensity projections of the same number of stacks for all cells, to create binarized masks of the apical area defined by the Arm marker. For the detection of the Kkv puncta and the subsequent creation of a second binary image, we used the "Find Maxima" function of the Fiji software, with the output set to Maxima with Tolerance. The two binary images created were used as an input for the Spatial Statistics 2D/3D Fiji plugin [26,28], and the parameters were set to 10,000 number of points for the F-Function, 200 random point pattern generation for the average CDF and SDI, hard-core distance of 0.08 to 0.18, and confidence limit for the CDF at 5%. For the calculation of the NND, the binary images of the Kkv spots created previously were used as an input for the Nearest Neighbor Distances Calculation with ImageJ plugin of the Fiji software. To achieve better object detection, the binary images created for salivary gland cells, resulted from preprocessed stacks by background subtraction and application of Maximum filter. The masks used

for the Spatial Statistics 2D/3D Fiji plugin were created after applying robust automatic thresh-old selection, the "open" function of the EDM Binary operations of the BioVoxxel toolbox and the Watershed segmentation tool. The values obtained can be found in the S1 Data. For the analysis of the results obtained, the Kolmogorov–Smirnov test was used to assess the sample distribution within the populations.

## Coimmunoprecipitation assay

Assays were performed with extracts prepared from *Drosophila* embryos that were lysed in RIPA buffer (50 mM Tris–HCl (pH 8),150 mM NaCl, 0.1% SDS, 0.5% sodium deoxycho-late,1% Triton X-100, 1 mM PMSF, and protease inhibitors (cOmplete Tablets, Roche)). Extracts were immunoprecipitated using anti-Kkv antibodies or a mock antibody (anti-CP190), followed by incubation with Protein G Dynabeads (Invitrogen). Immunoprecipitates were washed with RIPA buffer and analysed by western blot using anti-Kkv or anti-Reb anti-bodies and the Immobilon ECL reagent (Millipore). Original uncropped blots can be found in the S1 Raw Images.

## Supporting information

**S1 Fig. Effects of the coexpression of Reb and GFP-Kkv expression.** All images correspond to projections of confocal sections. (**A**-**D**) At early stages, overexpressed GFP-Kkv accumulates apically (white arrow in **B**) and in intracellular punctae (blue arrow in **B**), as endogenous Kkv (white and blue arrows in **A**), but also in the whole cell. At later stages, GFP-Kkv shows a pat-tern in stripes that corresponds to the taenidial folds (inset in **D**), in a comparable pattern to the endogenous Kkv (inset in **C**). Endogenous Kkv at late stages localises mainly apical and almost no intracellular punctae are detected (**C**). GFP-Kkv also localises mainly apical, but in addition, Kkv intracellular punctae are also detected (blue arrow in **D**). (**E**-**H**) In trachea, the simultaneous overexpression of *reb* and *GFP-kkv* anticipates chitin deposition (compare **E** and **F**). At later stages, this results in different morphogenetic defects like short and straight tubes and defects in branch fusion (compare **H** and **G**). (**I**) In salivary glands, the coexpression of *reb* and *GFP-kkv* promotes accumulation of chitin in the lumen. Scale bars: 10 μm.
(TIF)

**S2 Fig. Effects of the expression of Exp/Reb and GFP-Kkv.** All images are projections of con-focal sections. (**A**, **B**) Overexpressed full-length Reb localises mainly apically in trachea (**A**) and in salivary glands (**B**). (**C**) The simultaneous overexpression of *MH2-exp* and *GFP-kkv* does not rescue the absence of extracellular chitin deposition, and it produces intracellular chi-tin vesicles (pink arrowheads). (**D**) Endogenous Exp localises mainly apically in trachea, although a bit of the protein can be detected intracellularly. (**E**) Example of a salivary gland used to quantify the number of GFP-Kkv vesicles in Fig 5I. (**E'**) A mask to count the vesicles is generated by substracting background and the apical membrane region. Scale bars: 10 μm.
(TIF)

**S3 Fig. Summary of described phenotypes.** Summary of phenotypes of the different UAS constructs in wild-type (wt), *exp reb* mutant, and *kkv* mutant embryos in different overexpres-sion conditions. The phenotypes observed are indicated in black, and light grey indicates absence of the phenotype
(TIF)

**S4 Fig.** (**A**) Projection of confocal sections of salivary glands shows that Kkv is normally expressed in salivary glands and accumulates apically. (**B**-**D**) Single confocal sections of tra-chea at early stages. Single chitin punctae do not colocalise with deacetylases or Gasp. (**E**, **F**)

Single confocal sections of salivary glands expressing *serp* and *verm*. Single chitin punctae do not colocalise with deacetylases. (**G**) Frames from live imaging movie 2 show that common GFP-Kkv and chitin punctae (yellow arrow) can separate from each other; many GFP-Kkv (green arrow) and chitin puncta (red arrow) do not colocalise. (**H**) The luminal GFP-Kkv punctae are labelled by GFP and Kkv. Scale bars **A**, **E**, **F**: 5 μm; **C-D**, **G**: 10 μm.
(TIF)

**S5 Fig. Analysis of Kkv apical distribution in salivary glands and Co-IP.** (**A**, **B**) Kkv distribution on the apical surface of a control (**A**) and an embryo expressing *reb* in salivary glands (*fkhGal4>UAS-reb*, **B**) and zoomed images for Kkv punctae (magenta) on the apical cell area marked by arm (green) of a control (**C**) and a *fkhGal4>UAS-reb* (**D**) embryo. The corresponding observed F function for the control (**E**) and the *fkhGal4>UAS-reb* (**F**) are displayed within and below the reference simulated random distributions (black) and the 95% confidence interval (light gray), respectively, indicating a random spatial pattern for the control and a tendency towards the formation of aggregates for the *reb* ectopic expression. (**G**) SDI histogram for the F-Function of the control (blue) and the *fkhGal4>UAS-reb* (magenta) samples. A significant difference between the frequency distributions for each group of individuals has been observed. (Kolmogorov–Smirnov D = 0.6667, $p < 0.01$) (**H**) Frequency distribution histograms for the Nearest Neighbour Distances (NNDs) between Kkv punctae in control (blue) and *fkhGal4>UAS-reb* samples. The distribution of values between the two groups is found significantly different (Kolmogorov–Smirnov D = 0.1463, $p < 0.005$). All images are projections of confocal sections, of super-resolution microscopy. The underlying data for quantifications can be found in the S1 Data. (**J**, **K**) Western blot using αKkv (**J**, two different exposure times are shown) or αReb (**K**) of embryo extracts that were subjected to immunoprecipitation with αKkv or an unrelated antibody (mock). Input correspond to 7.5% of the immunoprecipitated material. The position of MW markers (in kDa) is indicated. Scale bars **A**, **B**: 5 μm; **C**, **D**: 2 μm.
(TIF)

**S1 Movie. GFP-kkv vesicles and chitin punctae.** Salivary gland of a stage 15 embryo carrying *GFP-kkv*, *ChtVisTomato*, and *exp^{ΔMH2}* visualised from a lateral view using Drangofly 505 (Andor) with 63× oil objective and a 2× zoom. Images were taken every 3 s in one single Z-stack during 2 min. The movie shows a chitin particle detaching from a GFP-kkv vesicle.
(MOV)

**S2 Movie. GFP-kkv vesicles and chitin punctae.** Salivary glands of a stage 15 embryos carrying *GFP-kkv*, *ChtVisTomato*, and *exp^{ΔMH2}* visualised from a lateral view using Drangofly 505 (Andor) with 63× oil objective and a 2× zoom. Images were taken every 3 s in one single Z-stack during 1 min and 30 s. The movie shows a chitin particle detaching from a GFP-kkv vesicle.
(MOV)

**S1 Data. Original quantification data obtained with Fiji software for main figures and supporting information are provided as an excel file.** Mean, standard deviation, standard error of mean, and details of statistical tests are indicated for each set of data analysed by GraphPad Prism 9.
(XLSX)

**S1 Raw Images. Original uncropped western blots for S5J and S5K Fig.** The excerpted portion of the immunoblot shown in the figures is highlighted by a black box.
(PDF)

## Acknowledgments

We thank E. Rebollo from the IBMB-PCB Molecular Imaging Platform for imaging and N. Giakoumakis and S. Tosi from IRB-Advanced Digital Microscopy for technical help and discussions on quantifications. We thank N. Martín for technical help and generation of Kkv antibody. We acknowledge the Bloomington Stock Centre and the Developmental Studies Hybridoma Bank for fly lines and antibodies. We thank T. Tanaka, S. Hayashi, and D. Andrew for kindly providing flies and antibodies. Thanks also go to the members of the Llimargas and Casanova labs for helpful discussions, and to J. Casanova and M. Furriols for critical reading of the manuscript.

## Author Contributions

**Conceptualization:** Ettore De Giorgio, Marta Llimargas.

**Formal analysis:** Ettore De Giorgio, Panagiotis Giannios, Marta Llimargas.

**Funding acquisition:** Marta Llimargas.

**Investigation:** Ettore De Giorgio, Panagiotis Giannios, M. Lluisa Espinàs, Marta Llimargas.

**Methodology:** Ettore De Giorgio, Panagiotis Giannios.

**Supervision:** Marta Llimargas.

**Writing – original draft:** Marta Llimargas.

**Writing – review & editing:** Ettore De Giorgio, Panagiotis Giannios, M. Lluisa Espinàs, Marta Llimargas.

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
