## [Editor Report · Decision Letter 0]

31 Oct 2022

Dear Marta, 

Thank you for submitting the revision of your Review Commons manuscript entitled "Dissecting the roles of Expansion/Rebuf and the chitin synthase Krotzkopf Verkehrt in chitin deposition in Drosophila" for consideration as a Research Article by PLOS Biology.

The revision has now been evaluated by the PLOS Biology editorial staff as well as by an academic editor with relevant expertise and I am writing to let you know that we would like to send back to the original reviewers.

However, before we can send your manuscript to the reviewers, we need you to complete your submission by providing the metadata that is required for full assessment. To this end, please login to Editorial Manager where you will find the paper in the 'Submissions Needing Revisions' folder on your homepage. Please click 'Revise Submission' from the Action Links and complete all additional questions in the submission questionnaire.

Once your full submission is complete, your paper will undergo a series of checks in preparation for peer review. After your manuscript has passed the checks it will be sent out for review. To provide the metadata for your submission, please Login to Editorial Manager (https://www.editorialmanager.com/pbiology) within two working days, i.e. by Nov 02 2022 11:59PM.

Kind regards,

Ines

--

Ines Alvarez-Garcia, PhD

Senior Editor

PLOS Biology

---

## [Decision Letter · Decision Letter 1]

2 Dec 2022

Dear Marta,

Thank you for your patience while we considered your revised manuscript entitled "Dissecting the roles of Expansion/Rebuf and the chitin synthase Krotzkopf Verkehrt in chitin deposition in Drosophila" for publication as a Research Article at PLOS Biology. This revised version of your manuscript, which we received from Review Commons, has been evaluated by the PLOS Biology editors, an the Academic Editor and the original three reviewers.

Based on the reviews (attached below), we are likely to accept this manuscript for publication, provided you satisfactorily address the remaining minor point raised by Reviewer 1. Please also make sure to address the data and other policy-related requests stated below.

In addition, we would like you to consider a suggestion to improve the title by simplifying it a bit (feel free to suggest something else if you don't like it):

"Chitin polymerisation and translocation to the extracellular space are uncoupled in Drosophila"

Please also define better what Expansion and Rebuf are in the abstract.

We expect to receive your revised manuscript within two weeks. 

*Published Peer Review History*

*Press*

Best wishes,

Ines

--

Ines Alvarez-Garcia, PhD

Senior Editor

PLOS Biology

Fig. 2G; Fig. 3O; Fig. 5I; Fig. 6M, N, N’, P, Q; Fig. S5E-H

We require the original, uncropped and minimally adjusted images supporting all blot and gel results reported in an article's figures or Supporting Information files. We will require these files before a manuscript can be accepted so please prepare and upload them now. Please carefully read our guidelines for how to prepare and upload this data: https://journals.plos.org/plosbiology/s/figures#loc-blot-and-gel-reporting-requirements

Reviewers' comments:

Rev. 1:

This is a revised manuscript and my previous comments have essentially been addressed successfully.

There is a minor comment regarding the expression pattern of endogenous KKV and GFP-KKV.

Figure S1. (C) In wild type embryos, endogenous KKV is apically localized, almost no intracellular KKV is observed. (d) in btl> GFP-KKV, in addition to apical localization, vesicular KKV-GFP is clearly observed.

These intracellular KKV-GFP vesicles might not contribute significantly to intracellular chitin in exp reb mutants, however, this observation should be mentioned.

Rev. 2: Matthias Behr – note that this reviewer has signed his review

Well done.

Rev. 3:

This is a revised version of a previously reviewed paper. The authors have done considerable to work both experimentally and in textual and figure revisions to address the reviewer comments, which they have done successfully. The revision has significantly strengthened the paper.

The manuscript is now ready for publication.

---

## [Editor Report · Decision Letter 2]

22 Dec 2022

Dear Dr Llimargas,

Thank you for the submission of your revised Research Article entitled "A dynamic interplay between chitin synthase and the proteins Expansion/Rebuf reveals that chitin polymerisation and translocation are uncoupled in Drosophila" for publication in PLOS Biology. On behalf of my colleagues and the Academic Editor, Emma Rawlins, I am delighted to say that we can in principle accept your manuscript for publication, provided you address any remaining formatting and reporting issues. These will be detailed in an email you should receive within 2-3 business days from our colleagues in the journal operations team; no action is required from you until then. Please note that we will not be able to formally accept your manuscript and schedule it for publication until you have completed any requested changes.

PRESS

Sincerely, 

Ines

--

Ines Alvarez-Garcia, PhD

Senior Editor

PLOS Biology
